# Neutrophils promote CXCR3-dependent itch in the development of atopic dermatitis

Carolyn M Walsh[1†§], Rose Z Hill[1†§‡], Jamie Schwendinger-Schreck[1], Jacques Deguine[1], Emily C Brock[1], Natalie Kucirek[1], Ziad Rifi[1], Jessica Wei[2], Karsten Gronert[2], Rachel B Brem[3,4], Gregory M Barton[1]*, Diana M Bautista[1,5]*

[1]Department of Molecular and Cell Biology, University of California, Berkeley, Berkeley, United States; [2]Vision Science Program, School of Optometry, University of California, Berkeley, Berkeley, United States; [3]Department of Plant and Microbial Biology, University of California, Berkeley, Berkeley, United States; [4]Buck Institute for Research on Aging, Novato, United States; [5]Helen Wills Neuroscience Institute, University of California, Berkeley, Berkeley, United States

**Abstract** Chronic itch remains a highly prevalent disorder with limited treatment options. Most chronic itch diseases are thought to be driven by both the nervous and immune systems, but the fundamental molecular and cellular interactions that trigger the development of itch and the acute-to-chronic itch transition remain unknown. Here, we show that skin-infiltrating neutrophils are key initiators of itch in atopic dermatitis, the most prevalent chronic itch disorder. Neutrophil depletion significantly attenuated itch-evoked scratching in a mouse model of atopic dermatitis. Neutrophils were also required for several key hallmarks of chronic itch, including skin hyperinnervation, enhanced expression of itch signaling molecules, and upregulation of inflammatory cytokines, activity-induced genes, and markers of neuropathic itch. Finally, we demonstrate that neutrophils are required for induction of CXCL10, a ligand of the CXCR3 receptor that promotes itch via activation of sensory neurons, and we find that that CXCR3 antagonism attenuates chronic itch.

*For correspondence:
barton@berkeley.edu (GMB);
dbautista@berkeley.edu (DMB)

†These authors contributed equally to this work
§Author order was randomly determined by a coin flip

Present address: ‡The Scripps Research Institute, La Jolla, United States

Competing interests: The authors declare that no competing interests exist.

## Introduction

Chronic itch is a debilitating disorder that affects millions of people worldwide (*Matterne et al., 2011*; *Mollanazar et al., 2016*; *Dalgard et al., 2015*). It is a symptom of a number of skin diseases and systemic disorders, as well as a side effect of a growing list of medications. Like chronic pain, chronic itch can be a disease in and of itself (*Ständer and Steinhoff, 2002*; *Oaklander, 2011*; *Dhand and Aminoff, 2014*). Unlike acute itch, which can facilitate removal of crawling insects, parasites, or irritants, persistent scratching in chronic itch disorders has no discernable benefit; scratching damages skin, leading to secondary infection, disfiguring lesions, and exacerbation of disease severity (*Mollanazar et al., 2016*; *Yosipovitch and Papoiu, 2008*; *Ikoma et al., 2006*). The most common chronic itch disorder is atopic dermatitis (AD; commonly known as eczema), which affects fifteen million people in the United States alone (*Spergel and Paller, 2003*). Severe AD can trigger the atopic march, where chronic itch and inflammation progress to food allergy, allergic rhinitis, and asthma (*Spergel and Paller, 2003*; *Zheng et al., 2011*).

Little is known about the underlying mechanisms that drive chronic itch pathogenesis. As such, studies of human chronic itch disorders have sought to identify candidate mechanisms of disease progression. A number of studies have identified biomarkers and disease genes in itchy human AD lesions (*Ewald et al., 2017*; *Choy et al., 2012*; *Guttman-Yassky et al., 2009*; *Suárez-Fariñas et al., 2013*; *Jabbari et al., 2012*). Indeed, a recent study compared the transcriptomes of healthy skin to

**eLife digest** Chronic itch is a debilitating disorder that can last for months or years. Eczema, or atopic dermatitis, is the most common cause for chronic itch, affecting one in ten people worldwide. Many treatments for the condition are ineffective, and the exact cause of the disease is unknown, but many different types of cells are likely involved. These include skin cells and inflammation-promoting immune cells, as well as nerve cells that detect inflammation, relay itch and pain information to the brain, and regulate the immune system.

Learning more about how these cells interact in eczema may help scientists find better treatments for the condition. So far, a lot of research has focused on static 'snapshots' of mature eczema lesions from human skin or animal models. These studies have identified abnormalities in genes or cells, but have not revealed how these genes and cells interact over time to cause chronic itch and inflammation.

Now, Walsh et al. reveal that immune cells called neutrophils trigger chronic itch in eczema. The experiments involved mice with a condition that mimics eczema, and showed that removing the neutrophils in these mice alleviated their itching. They also showed that dramatic and rapid changes occur in the nervous system of mice suffering from the eczema-like condition. For example, excess nerves grow in the animals' damaged skin, genes in the nerves that detect sensations become hyperactive, and changes occur in the spinal cord that have been linked to nerve pain. When neutrophils are absent, these changes do not take place.

These findings show that neutrophils play a key role in chronic itch and inflammation in eczema. Drugs that target neutrophils, which are already used to treat other diseases, might help with chronic itch, but they would need to be tested before they can be used on people with eczema.

itchy and non-itchy skin from psoriasis and AD patients, revealing dramatic changes in expression of genes associated with cytokines, immune cells, epithelial cells, and sensory neurons (*Nattkemper et al., 2018*). However, due to the difficulty in staging lesion development and obtaining staged samples from patients, there is currently no temporal map of when individual molecules and cell types contribute to chronic itch pathogenesis. Furthermore, the use of human patient data does not allow for rigorous mechanistic study of how disease genes contribute to chronic itch. To this end, we used a well-characterized inducible animal model of itch to define where, when, and how these genes identified from patient data contribute to chronic itch pathogenesis.

We employed the MC903 mouse model of AD and the atopic march (*Dai et al., 2017*; *Li et al., 2009*; *Li et al., 2006*; *Zhang et al., 2009*; *Moosbrugger-Martinz et al., 2017*) to provide a framework within which to identify the molecules and cells that initiate the development of atopic itch. The MC903 model is ideal for our approach because of its highly reproducible phenotypes that closely resemble human AD and its ability to induce the development of lesions and scratching (*Li et al., 2009*; *Li et al., 2006*; *Zhang et al., 2009*; *Oetjen et al., 2017*; *Morita et al., 2015*; *Kim et al., 2019*). By contrast, it is difficult to synchronously time the development of lesions in commonly used genetic models of AD, such as filaggrin mutant mice or Nc/Nga mice. Another advantage of the MC903 model is that it displays collectively more hallmarks of human AD than any one particular genetic mouse model. For example, the commonly used IL-31$^{tg}$ overexpressor model (*Cevikbas et al., 2014*; *Meng et al., 2018*) lacks strong Th2 induction, (*Martel et al., 2017*) and itch behaviors have not yet been rigorously characterized in the keratinocyte-TSLP overexpressor model. As MC903 is widely used to study the chronic phase of AD, we hypothesized that MC903 could also be used to define the early mechanisms underlying the development of chronic itch, beginning with healthy skin. We performed RNA-seq of skin at key time points in the model. We complemented this approach with measurements of itch behavior and immune cell infiltration. The primary goal of our study was to identify the inciting molecules and cell types driving development of chronic itch. To that end, we show that infiltration of neutrophils into skin is required for development of chronic itch. Additionally, we demonstrate that neutrophils direct early hyperinnervation of skin, and the upregulation of itch signaling molecules and activity-induced genes in sensory neurons. Finally, we identify CXCL10/CXCR3 signaling as a key link between infiltrating neutrophils and sensory neurons that drives itch behaviors.

## Results

### MC903 triggers rapid changes in expression of skin barrier, epithelial cell-derived cytokine, and axon guidance genes

Although a variety of AD- and chronic itch-associated genes have been identified, when and how they contribute to disease pathogenesis is unclear. Using RNA-seq of MC903-treated skin, we observed distinct temporal patterns by which these classes of genes are differentially expressed across the first eight days of the model (*Figure 1A–B*, *Figure 1—figure supplement 1A*). Overall, we found that 62% of genes from a recent study of human chronic itch lesions (*Nattkemper et al., 2018*) (*Figure 1—figure supplement 1A*) and 67% of AD-related genes (*Figure 1B*) were significantly changed for at least one of the time points examined, suggesting that the MC903 mouse model recapitulates many key transcriptional changes occuring in human chronic itch and AD. MC903 dramatically alters the transcriptional profile of keratinocytes by derepressing genomic loci under the control of the Vitamin D Receptor. In line with rapid changes in transcription, proteases (*Klk6*, *Klk13*, among others) and skin barrier genes (*Cdhr1*) changed as early as six hours after the first treatment, before mice begin scratching (*Figure 1B*). Increased protease activity in AD skin is thought to promote breakdown of the epidermal barrier and release of inflammatory cytokines from keratinocytes (*Rattenholl and Steinhoff, 2003*; *Yosipovitch, 2004*). One such cytokine, thymic stromal lymphopoetin (TSLP) is a key inducer of the Type two immune response, which is characteristic of human AD and the MC903 model, via signaling in CD4$^+$ T cells, basophils, and other immune cells (*Li et al., 2006*; *Zhang et al., 2009*; *Briot et al., 2010*; *Demehri et al., 2009*; *Gao et al., 2010*; *Kim et al., 2013*). Beginning at day two, before any significant itch-evoked scratching (*Figure 1C*), immune cell infiltration (*Figure 1E–G*, *Figure 1—figure supplements 3A*, *4A* and *5A–C*), or skin lesions (data not shown) (*Morita et al., 2015*) were observed, we saw increases in *Tslp*, as well as several other epithelial-derived cytokines, including the neutrophil chemoattractant genes *Cxcl1*, *Cxcl2*, *Cxcl3*, and *Cxcl5* (*Figure 1D*). To ask whether upregulation of these chemokine genes was dependent on protease activity, we treated human keratinocytes with the protease-activated receptor two agonist SLIGRL. SLIGRL treatment triggered increased expression of several of these chemokine genes, including *IL8*, the human ortholog of mouse *Cxcl1/Cxcl2*, and *CXCL2* (*Figure 1—figure supplement 6A*). These increases occurred after a few hours of exposure to SLIGRL, suggesting that increased protease activity can rapidly trigger increases in neutrophil chemoattractants in skin, similar to what we observe in MC903-treated mouse skin.

Unexpectedly, in the skin we observed early changes in a number of transcripts encoding neuronal outgrowth factors (*Ngf*, *Artn*) and axon pathfinding molecules (*Slit1*, *Sema3d*, *Sema3a*), some of which are directly implicated in chronic itch (*Hidaka et al., 2017*; *Kou et al., 2012*; *Tominaga and Takamori, 2013*; *Tominaga et al., 2007*; *Tominaga and Takamori, 2014*; *Figure 1—figure supplement 7A*), prior to when mice began scratching. We thus used immunohistochemistry (IHC) of whole-mount skin to examine innervation at this time point. We saw increased innervation of lesions at day two but not day one of the model (*Figure 1H–I*, *Figure 1—figure supplement 8A*). Our RNA-seq data showed elevation in skin CGRP transcript *Calca*, along with other markers of peptidergic nerve endings, specifically at day 2. Indeed, we saw an increase in CGRP$^+$ innervation of skin at day 2 (*Figure 1J*, *Figure 1—figure supplement 9A*), which suggests that elevation of neuronal transcripts in skin is due to hyperinnervation of peptidergic itch and/or pain fibers. The increased innervation was surprising because such changes had previously only been reported in mature lesions from human chronic itch patients (*Nattkemper et al., 2018*; *Haas et al., 2010*; *Kamo et al., 2013*; *Oaklander and Siegel, 2005*; *Schüttenhelm et al., 2015*; *Pereira et al., 2016*; *Tominaga et al., 2009*). Our findings suggest that early hyperinnervation is promoted by local signaling in the skin and is independent of the itch-scratch cycle.

### Neutrophils are the first immune cells to infiltrate AD skin

By day five, mice exhibited robust itch behaviors (*Figure 1C*) and stark changes in a number of AD disease genes (*Figure 1A–B*). For example, loss-of-function mutations in filaggrin (*FLG*) are a major risk factor for human eczema (*Palmer et al., 2006*; *Sandilands et al., 2007*). Interestingly, *Flg2* levels sharply decreased at day five. In parallel, we saw continued and significant elevation in neutrophil and basophil chemoattractant genes (*Cxcl1,2,3,5*, and *Tslp*, *Figure 1D*). Using flow cytometry, we

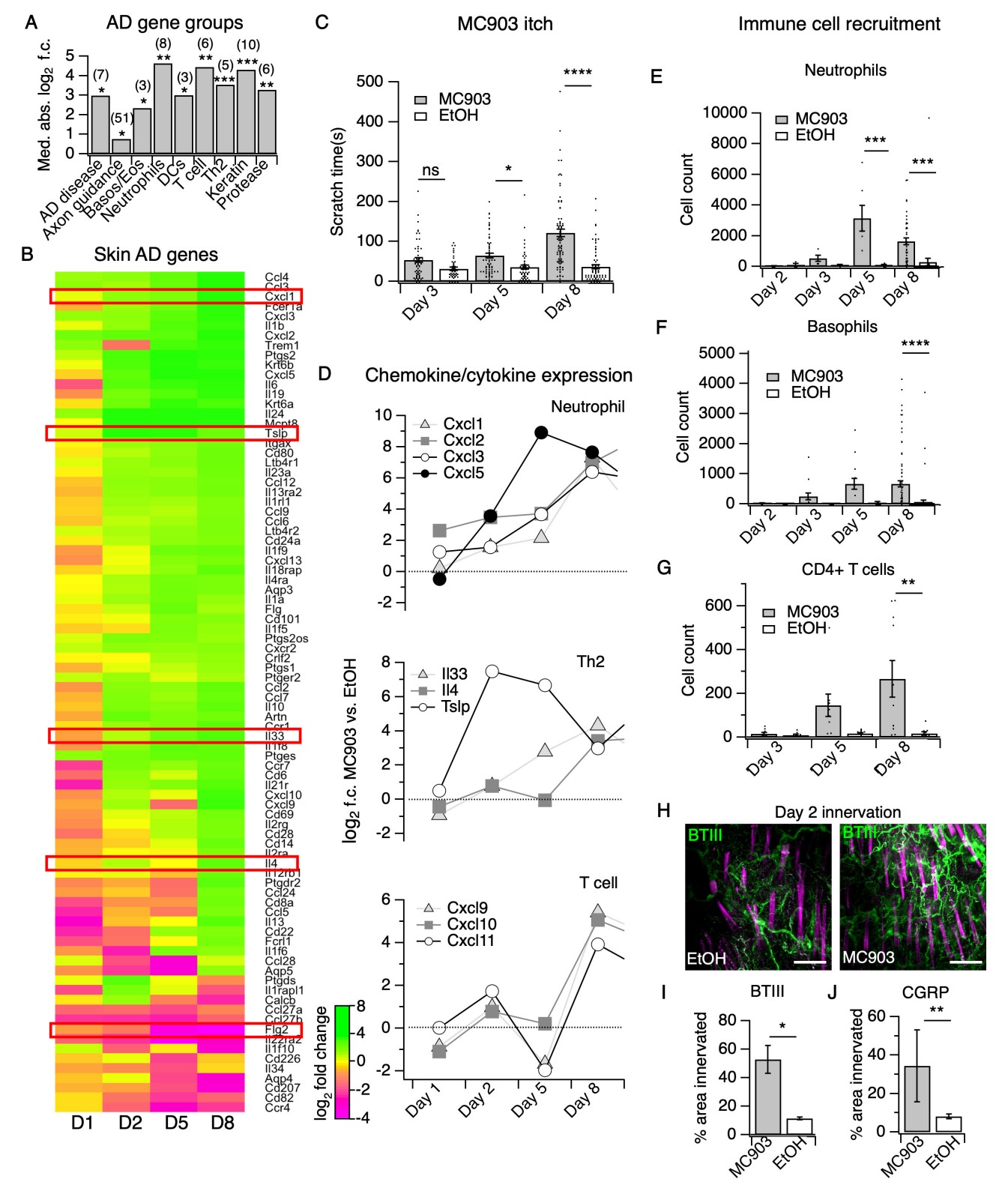

**Figure 1.** The MC903 model parallels the progression of human atopic disease and suggests a temporal sequence of AD pathogenesis. (**A**) Exact permutation test (10,000 iterations, see Materials and methods) for significance of mean absolute log2 fold change in gene expression at Day 8 (MC903 vs. ethanol) of custom-defined groups of genes for indicated categories (see **Figure 1—source data 1**). (**B**) Log2 fold change in gene expression (MC903 vs. ethanol) in mouse skin at indicated time points for key immune and mouse/human AD genes that were significantly differentially expressed

*Figure 1 continued on next page*

*Figure 1 continued*

for at least one time point in the MC903 model. Only genes from our initial list (see Materials and methods) differentially expressed at corrected p<0.05 and changing >2 fold between treatments for at least one condition are shown. Green bars = increased expression in MC903 relative to ethanol; magenta = decreased expression. Exact values and corrected *p*-values are reported in *Figure 1—source data 2* and *Source Data 1* Supplemental Data, respectively. D1 = 6 hr post-treatment; D2 = Day 2; D5 = Day 5; D8 = Day 8. (**C**) Scratching behavior of mice treated with MC903 or ethanol for indicated length of time (two-way ANOVA: ****$p_{interaction}$ <0.0001, F(2,409) = 13.25; Sidak's multiple comparisons: $p_{day\ 3}$ = 0.1309, n = 62,51 mice; *$p_{day\ 5}$ = 0.0171, n = 69,56 mice; ****$p_{day\ 8}$ < 0.0001, n = 92,85 mice). Exact values displayed in *Figure 1—source data 3*. (**D**) Log$_2$ fold change in gene expression of neutrophil chemoattractants (upper), Th2 cytokines (middle) and T cell chemoattractants (lower, from RNA-seq data). (**E**) Neutrophil counts in MC903- and ethanol-treated skin at indicated time points (two-way ANOVA: **$p_{treatment}$ = 0.0023, F(1,102) = 9.82; Sidak's multiple comparisons: $p_{day\ 2}$ > 0.999, n = 4,4 mice; $p_{day\ 3}$ = 0.9801, n = 5,5 mice; ***$p_{day\ 5}$ = 0.0003, n = 6,8 mice; ***$p_{day\ 8}$ = 0.0001, n = 40,38 mice). (**F**) Basophil counts in MC903- and ethanol-treated skin at indicated time points (two-way ANOVA: **$p_{treatment}$ = 0.0051, F(1,102) = 8.17; Sidak's multiple comparisons: $p_{day\ 2}$ > 0.999, n = 4,4 mice; $p_{day\ 3}$ = 0.8850, n = 5,5 mice; $p_{day\ 5}$ = 0.0606, n = 6,8 mice; ****$p_{day\ 8}$ < 0.0001, n = 40,38 mice). (**G**) CD4$^+$ T cell counts in MC903- and ethanol-treated skin at indicated time points (two-way ANOVA: **$p_{time}$ = 0.0042, F(1,44) = 9.10; $p_{day\ 3}$ = 0.9998, n = 8,6 mice; $p_{day\ 5}$ = 0.2223, n = 9,8 mice; **$p_{day\ 8}$ = 0.0021, n = 11,8 mice). Day 8 immune cell infiltrate represented as % of CD45$^+$ cells in *Figure 1—figure supplement 2A–B* (see *Supplementary file 3* for all experimental conditions). Exact values displayed in *Figure 1—source data 4* and representative FACS plots for myeloid and T cell gating shown in *Figure 1—figure supplement 3A* and *Figure 1—figure supplement 4A*. For *Figure 4E–G*, data from mice receiving i.p. injection of PBS (see *Figure 4*) in addition to MC903 or EtOH are also included. (**H**) (Upper and Lower) Representative maximum intensity Z-projections from immunohistochemistry (IHC) of whole-mount mouse skin on Day 2 of the MC903 model. Skin was stained with neuronal marker beta-tubulin III (BTIII; green). Hair follicle autofluorescence is visible in the magenta channel. Images were acquired on a confocal using a 20x water objective. (**I**) Quantification of innervation (see Materials and methods) of mouse skin as determined from BTIII staining (*p=0.012; two-tailed t-test (t = 3.114; df = 9); n = 7,4 images each from two mice per treatment). Day 1 IHC results as follows: 31.78 ± 18.39% (MC903) and 31.51 ± 16.43% (EtOH); p=0.988; two-tailed unpaired t-test; n = 6 images each from two mice per treatment. Exact values are reported in *Figure 1—source data 5*. (**J**) Quantification of CGRP$^+$ nerve fibers (see Materials and methods) in skin (**p=0.0083; two-tailed t-test (t = 2.868; df = 25); n = 15, 12 images from three mice per treatment). Exact values are reported in *Figure 1—source data 5*. Representative images in *Figure 1—figure supplement 9A*. The online version of this article includes the following source data and figure supplement(s) for figure 1:

**Source data 1.** Values displayed in the bar plot shown in *Figure 1A*.
**Source data 2.** Values displayed in the heat map shown in *Figure 1B*.
**Source data 3.** Values displayed in the bar plot shown in *Figure 1C*.
**Source data 4.** Values displayed in the bar plots shown in *Figure 1E–G* and *Figure 1—figure supplement 5A–C*.
**Source data 5.** Values displayed in the bar plots shown in *Figure 1I* and *Figure 1J*.
**Source data 6.** Values displayed in the heat map shown in *Figure 1—figure supplement 1A*.
**Source data 7.** Values displayed in the heat map shown in *Figure 1—figure supplement 6A*.
**Source data 8.** Values displayed in the heat map shown in *Figure 1—figure supplement 7A*.
**Source data 9.** Values displayed in the bar plot shown in *Figure 1—figure supplement 10A*.
**Figure supplement 1.** Expression of mouse and human itch genes.
**Figure supplement 2.** Immune cells represented as % of CD45$^+$ cells.
**Figure supplement 3.** Myeloid and granulocyte gating strategy.
**Figure supplement 4.** T cell gating strategy.
**Figure supplement 5.** Immune cell counts in MC903-treated skin.
**Figure supplement 6.** Protease receptor activation triggers rapid upregulation of neutrophil chemoattractant genes in human keratinocytes.
**Figure supplement 7.** Expression of neuronal genes and axon guidance molecules in skin.
**Figure supplement 8.** Method of image quantification for whole mount skin.
**Figure supplement 9.** Peptidergic fibers display hyperinnervation in MC903-treated skin.
**Figure supplement 10.** Inflammatory lipids in MC903-treated skin.

observed a number of infiltrating immune cells in the skin at day 5. Of these, we neutrophils were the most abundant immune cell subtype (*Figure 1E*, *Figure 1—figure supplement 3A*). It was not until day eight that we observed the classical AD-associated immune signature in the skin, (*Gittler et al., 2012*) with upregulation of *Il4*, *Il33* and other Th2-associated genes (*Figure 1B*, *Figure 1D*). We also observed increases in the T cell chemoattractant genes *Cxcl9*, *Cxcl10*, and *Cxcl11* (*Figure 1D*), which are thought to be hallmarks of chronic AD lesions in humans (*Oetjen and Kim, 2018*; *Mansouri and Guttman-Yassky, 2015*). Neutrophils and a number of other immune cells that started to infiltrate on day five were robustly elevated in skin by day eight, including basophils (*Figure 1F*), CD4$^+$ T cells (*Figure 1G*, *Figure 1—figure supplement 4A*), eosinophils (*Figure 1—figure supplement 5C*), and mast cells (*Figure 1—figure supplement 5B*), but not inflammatory monocytes (*Figure 1—figure supplement 5A*).

CD4$^+$ T cells are ubiquitous in mature human AD lesions (*Guttman-Yassky and Krueger, 2017*) and promote chronic AD itch and inflammation. More specifically, they play a key role in IL4Rα-dependent sensitization of pruriceptors in the second week of the MC903 model (*Oetjen et al., 2017*). Thus, we were quite surprised to find that itch behaviors preceded significant CD4$^+$ T cell infiltration. Therefore, neutrophils drew our attention as potential early mediators of MC903 itch. While neutrophil infiltration is a hallmark of acute inflammation, it remains unclear whether neutrophils contribute to the pathogenesis of chronic itch. Moreover, neutrophils release known pruritogens, including proteases, reactive oxygen species, and/or histamine, inflammatory lipids, and cytokines that sensitize and/or activate pruriceptors (*Dong and Dong, 2018*; *Hashimoto et al., 2018*). Increased levels of the prostaglandin PGE$_2$ and the neutrophil-specific leukotriene LTB$_4$ have also been reported in skin of AD patients (*Fogh et al., 1989*). Indeed, by mass spectrometry, we observed increases in several of these inflammatory lipids, PGD$_2$ and PGE$_2$, as well as LTB$_4$ and its precursor 5-HETE (*Figure 1—figure supplement 10A*) in MC903-treated skin, implicating neutrophils in driving AD itch and inflammation. Thus, we next tested the requirement of neutrophils to itch in the MC903 model.

## Neutrophils are required for early itch behaviors in the MC903 model of AD

We first asked whether neutrophils, the most abundant population of infiltrating immune cells in this chronic itch model, were required for MC903-evoked itch. Systemic depletion of neutrophils using daily injections of an anti-Gr1 (aGr1) antibody (*Ghasemlou et al., 2015*; *Sivick et al., 2014*) dramatically attenuated itch-evoked scratching through the first eight days of the model (*Figure 2A*). Consistent with a key role for neutrophils in driving chronic itch, our depletion strategy significantly and selectively reduced circulating and skin infiltrating neutrophils on days five and eight, days on which control, but not depleted mice, scratched robustly (*Figure 2B*; *Figure 2—figure supplement 1A–C*). In contrast, basophils and CD4$^+$ T cells continued to infiltrate the skin following aGr1 treatment (*Figure 2C–D*), suggesting that these cells are not required for early MC903 itch.

We next used the cheek model of acute itch (*Shimada and LaMotte, 2008*) to ask whether neutrophil recruitment is sufficient to trigger scratching behaviors. As expected, we observed significant and selective recruitment of neutrophils to cheek skin within 15 min after CXCL1 injection (*Figure 2—figure supplement 2A–B*). CXCL1 injection also triggered robust scratching behaviors (*Figure 2E*) on a similar time course to neutrophil infiltration (*Figure 2—figure supplement 2B*). Thus, we next acutely depleted neutrophils with aGr1 to determine whether neutrophils were required for CXCL1-evoked acute itch. Indeed, aGr1-treatment rapidly reduced circulating neutrophils (*Figure 2—figure supplement 2C*) and resulted in a dramatic loss of CXCL1-evoked itch behaviors (*Figure 2C*). This effect was specific to neutrophil-induced itch, as injection of chloroquine, a pruritogen that directly activates pruriceptors to trigger itch, still triggered robust scratching in aGr1-treated animals (*Figure 2—figure supplement 3A*). Given that CXCL1 has been shown to directly excite and/or sensitize sensory neurons, (*Deftu et al., 2017*; *Deftu et al., 2018*) it is possible that the mechanism by which CXCL1 elicits itch may also involve neuronal pathways. However, our results show that CXCL1-mediated neutrophil infiltration is sufficient to drive acute itch behaviors, and that neutrophils are necessary for itch in the MC903 model.

We also examined MC903-evoked itch behaviors in mice deficient in *Crlf2*, the gene encoding the TSLP Receptor (TSLPR KO mice; *Carpino et al., 2004*). TSLPR is expressed by both immune cells and sensory neurons and is a key mediator of AD in humans and in mouse models (*Li et al., 2009*; *Li et al., 2006*; *Zhang et al., 2009*; *Demehri et al., 2009*; *Briot et al., 2009*). Surprisingly, MC903-treated TSLPR KO mice displayed robust scratching behaviors through the first eight days of the model (*Figure 2F*). In contrast to our results in aGr1-injected mice, TSLPR KO mice displayed robust neutrophil infiltration (*Figure 2G*), but completely lacked basophil and CD4$^+$ T cell infiltration into the skin (*Figure 2H–I*, *Figure 2—figure supplement 4A*), and additionally displayed a reduction in mast cells (*Figure 2—figure supplement 4A*). These results suggest that basophils and CD4$^+$ T cells are not required for early itch and further support an inciting role for neutrophils. Previous studies have shown that TSLP drives the expression of Type two cytokines and related immune cells that promote itch and inflammation in mature AD skin lesions (*Li et al., 2009*; *Li et al., 2006*; *Zhang et al., 2009*; *Demehri et al., 2009*; *Briot et al., 2009*). Consistent with a later role for TSLP signaling in AD, we did observe a significant reduction in itch-evoked scratching in TSLPR KO mice

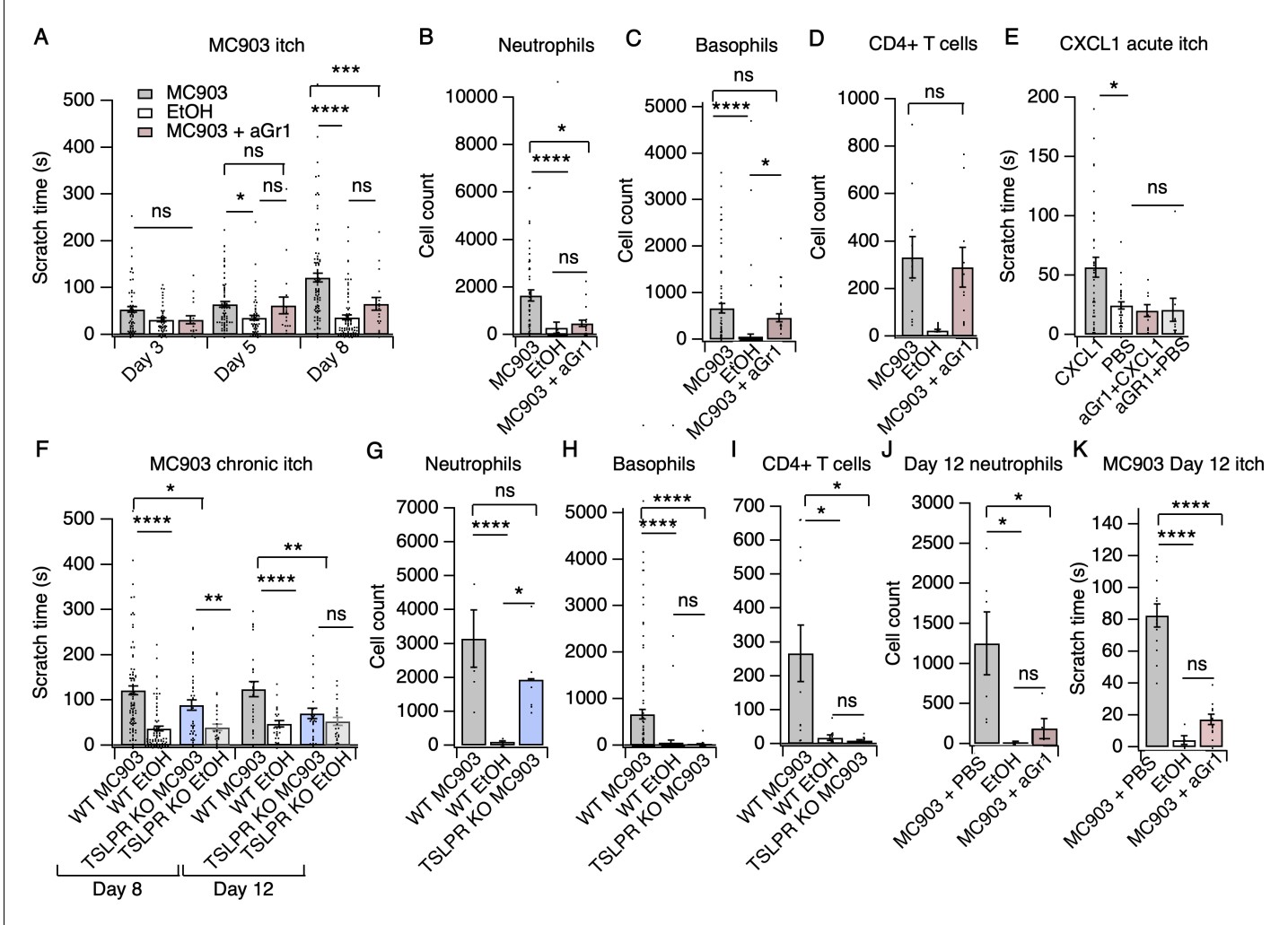

**Figure 2.** Neutrophils are necessary and sufficient for itch behaviors. (**A**) Scratching behavior of uninjected and PBS-injected mice (combined) and aGr1-injected mice treated with MC903 or ethanol for indicated length of time (two-way ANOVA: ****$p_{interaction}$ <0.0001, F(4,447) = 7.16; Tukey's multiple comparisons: $p_{day\ 3\ MC903\ vs.\ EtOH}$ = 0.1111 n = 62,51,17 mice; *$p_{day\ 5\ MC903\ vs.\ EtOH}$ = 0.0154, $p_{day\ 5\ MC903\ vs.\ aGr1}$ = 0.9854, $p_{day\ 5\ aGr1\ vs.}$ $_{EtOH}$ = 0.2267, n = 69,56,17 mice; ****$p_{day\ 8\ MC903\ vs.\ EtOH}$ <0.0001, ***$p_{day\ 8\ MC903\ vs.\ aGr1}$ = 0.0007, $p_{day\ 8\ aGr1\ vs.\ EtOH}$ = 0.1543, n = 92,85,17 mice). (**B**) Neutrophil count from cheek skin of uninjected/PBS-injected MC903- and ethanol-treated, and aGr1-injected MC903-treated mice on day 8 (one-way ANOVA: ****p<0.0001, F(2,92) = 10.59; Tukey's multiple comparisons: ****$p_{MC903\ vs.\ EtOH}$ <0.00001, n = 40,38 mice; *$p_{MC903\ vs.\ aGr1\ MC903}$ = 0.0109, n = 40,17 mice; $p_{aGr1\ vs.\ EtOH}$ = 0.8859, n = 38,17 mice). (**C**) Basophil count from cheek skin of uninjected/PBS-injected MC903- and ethanol-treated, and aGr1-injected MC903-treated mice on day 8 (one-way ANOVA: ****p=0.0001, F(2,92) = 14.61; Tukey's multiple comparisons: $p_{MC903\ vs.\ aGr1}$ $_{MC903}$ = 0.3217, n = 40,17 mice, ****$p_{MC903\ vs.\ EtOH}$ <0.0001, n = 40,38 mice, *$p_{aGr1\ MC903\ vs.\ EtOH}$ = 0.0204, n = 17,38 mice). (**D**) CD4+ T cell count from cheek skin of PBS-injected MC903- and ethanol-treated, and aGr1-injected MC903-treated mice on day 8 (two-way ANOVA: **$p_{treatment}$ = 0.0035, F (1,35) = 9.82; Holm-Sidak multiple comparisons for PBS versus aGr1: $p_{MC903}$ = 0.8878, n = 9,11 mice; $p_{EtOH}$ = 0.5201, n = 8,9 mice). Control MC903 and EtOH data from *Figure 2B–C* are also displayed in *Figure 1*. Exact values displayed for *Figure 2A–D* in *Figure 2—source data 1*. (**E**) Scratching behavior of mice immediately after injection of 1 μg CXCL1 or PBS (s.c. cheek). For neutrophil-depletion experiments, mice received 250 μg anti-Gr1 (aGr1) 20 hr prior to cheek injection of CXCL1 or PBS (one-way ANOVA: ****p<0.0001, F(4,88) = 75.53; Tukey's multiple comparisons: *$p_{CXCL1\ vs.}$ $_{PBS}$ = 0.0126, n = 36,31 mice; $p_{aGr1-CXCL1\ vs.\ aGr1-PBS}$ > 0.9999, n = 10,10 mice; $p_{aGr1-CXCL1\ vs.\ PBS}$ = 0.9986, n = 10,31 mice). Exact values displayed in *Figure 2—source data 2*. (**F**) Scratching behavior of WT and TSLPR KO (TSLPR KO) mice treated with MC903 or ethanol for indicated length of time (two-way ANOVA: ****$p_{interaction}$ <0.0001, F(9,657) = 4.93; Tukey's multiple comparisons: ****$p_{day\ 8\ WT\ MC903\ vs.\ EtOH}$ <0.0001, *$p_{day\ 8\ WT\ MC903\ vs.\ KO}$ $_{MC903}$ = 0.0194, **$p_{day\ 8\ KO\ MC903\ vs.\ KO\ EtOH}$ = 0.0039, n = 92,85,36,26 mice; ****$p_{day\ 12\ WT\ MC903\ vs.\ EtOH}$ <0.0001, **$p_{day\ 12\ WT\ MC903\ vs.\ KO\ MC903}$ = 0.0028, $p_{day\ 12\ KO\ MC903\ vs.\ KO\ EtOH}$ = 0.7061, n = 26,26,27,23 mice). (**G**) Neutrophil count from cheek skin of wild-type MC903- and ethanol-treated, and TSLPR KO MC903-treated mice on day 5 (two-way ANOVA: **$p_{genotype}$ = 0.0025, F(2,125) = 6.28; Tukey's multiple comparisons: ****$p_{day\ 5\ WT\ MC903\ vs.\ WT}$ $_{EtOH}$ <0.0001, n = 6,8 mice; $p_{day\ 5\ WT\ MC903\ vs.\ KO\ MC903}$ = 0.2198, n = 6,6 mice; *$p_{day\ 5\ WT\ EtOH\ vs.\ KO\ MC903}$ = 0.0212, n = 8,6 mice). (**H**) Basophil count from cheek skin of wild-type MC903- and ethanol-treated, and TSLPR KO MC903-treated mice on day 8 (two-way ANOVA: **$p_{genotype}$ = 0.0003, F(2,117) = 8.87; Tukey's multiple comparisons: ****$p_{day\ 8\ WT\ MC903\ vs.\ WT\ EtOH}$ <0.0001, n = 40,38 mice; ****$p_{day\ 8\ WT\ MC903\ vs.\ KO\ MC903}$ <0.0001, n = 40,15 mice; $p_{day\ 8\ WT\ EtOH\ vs.\ KO\ MC903}$ = 0.9519, n = 38,15 mice). See also *Figure 2—figure supplement 5A*. For *Figure 2G–H*, data from days 3, 5, and 8 are

*Figure 2 continued on next page*

Figure 2 continued

presented in *Figure 2—source data 3*. (I) CD4$^+$ T cell count from cheek skin of wild-type MC903- and ethanol-treated, and TSLPR KO MC903-treated mice on day 8 (one-way ANOVA: **p=0.0053, F(2,24) = 6.564; Tukey's multiple comparisons: *$p_{WT\ MC903\ vs.\ WT\ EtOH}$ = 0.0163, n = 11,8 mice; *$p_{MC903\ vs.\ KO\ MC903}$ = 0.0130, n = 11,8 mice; $p_{WT\ EtOH\ vs.\ KO\ MC903}$ = 0.9953, n = 8,8 mice). Wild-type MC903 and EtOH data from 2 F-H are also displayed in *Figure 1*. Exact values for *Figure 2F–I* displayed in *Figure 2—source data 3*. (J) Neutrophil count from cheek skin of wild-type MC903- and ethanol-treated mice on day 12 of the MC903 model. MC903-treated animals received daily i.p. injections of 250 μg aGr1 antibody or PBS (250 μL) on days 8–11 of the model (one-way ANOVA: *p=0.01, F(2,13) = 6.69; Tukey's multiple comparisons: *$p_{MC903-PBS\ vs.\ EtOH}$ = 0.0141, n = 6,5 mice; *$p_{MC903-PBS\ vs.\ MC903-aGr1}$ = 0.10330, n = 6,5 mice; $p_{MC903-aGr1\ vs.\ EtOH}$ = 0.9005, n = 5,5 mice). (K) Time spent scratching over a thirty minute interval for wild-type MC903- and ethanol-treated mice on day 12 of the MC903 model. MC903-treated animals received daily i.p. injections of 250 μg aGr1 antibody or PBS (250 μL) on days 8–11 of the model (one-way ANOVA: ****p<0.0001, F(2,26) = 53.1; Tukey's multiple comparisons: ****$p_{MC903-PBS\ vs.\ EtOH}$ <0.0001, n = 12,5 mice; ****$p_{MC903-PBS\ vs.\ MC903-aGr1}$ < 0.0001, n = 12,12 mice; $p_{MC903-aGr1\ vs.\ EtOH}$ = 0.3734, n = 12,5 mice). Values from bar plots are reported in *Figure 2—source data 5*.

The online version of this article includes the following source data and figure supplement(s) for figure 2:

**Source data 1.** Values displayed in bar plots shown in *Figure 2A–D*.
**Source data 2.** Values displayed in the bar plots shown in *Figure 2E* and Figs.
**Source data 3.** Values displayed in the bar plots shown in *Figure 2F–I* and *Figure 2—figure supplement 4A–B*.
**Source data 4.** Values used to generate the line plots shown in *Figure 2—figure supplement 1C*.
**Source data 5.** Values displayed in the bar plots shown in *Figure 2J–K*.
**Source data 6.** Values displayed in the bar plots in *Figure 2—figure supplement 5A*.
**Figure supplement 1.** aGr1 treatment preferentially depletes neutrophils.
**Figure supplement 2.** CXCL1 rapidly and selectively recruits neutrophils to skin.
**Figure supplement 3.** Neutrophil depletion does not affect chloroquine-evoked itch.
**Figure supplement 4.** Loss of TSLPR reduces skin basophil and mast cell numbers in the first week of AD development.
**Figure supplement 5.** Neutrophils robustly infiltrate the skin in the DNFB mouse model of atopic dermatitis.

in the second week of the model (*Figure 2F*). Thus, our data support a model in which neutrophils are necessary for initiation of AD and itch behaviors early in the development of AD, whereas TSLPR signaling mediates the recruitment of basophils and CD4$^+$ T cells to promote later stage itch and chronic inflammation.

The incomplete loss of itch behaviors on day 12 in the TSLPR KO animals (*Figure 2F*) raised the question of whether neutrophils might also contribute to itch during the second week of the MC903 model. To directly answer this question, we measured neutrophil infiltration and itch-evoked scratching on day 12 in mice that received either aGr1 or PBS on days 8–11 of the model to selectively deplete neutrophils solely during the second week. Neutrophil depletion in the second week with aGr1 robustly decreased skin-infiltrating neutrophils (*Figure 2J*), and substantially reduced scratching behaviors at day 12 (*Figure 2K*), supporting a role for neutrophils in chronic itch. Interestingly, we observed a 79% mean reduction in time spent scratching after neutrophil depletion at day 12, whereas loss of TSLPR effected a 44% reduction in time spent scratching. We speculate that neutrophils and TSLP signaling comprise independent mechanisms that together account for the majority of AD itch. In order to ascertain whether neutrophils could be salient players in other models of AD, and not just MC903, we measured neutrophil infiltration into ear skin in the 1-fluoro-2,4-dinitrobenzene (DNFB) model of atopic dermatitis, which relies on hapten-induced sensitization to drive increased IgE, mixed Th1/Th2 cytokine response, skin thickening, inflammation, and robust scratching behaviors in mice (*Zhang et al., 2015*; *Kitamura et al., 2018*; *Solinski et al., 2019a*). Indeed, neutrophils also infiltrated DNFB- but not vehicle-treated skin (*Figure 2—figure supplement 5A*). Taken together, these observations are complementary to published datasets showing evidence for neutrophil chemokines and transcripts in human AD lesions (*Ewald et al., 2017*; *Choy et al., 2012*; *Guttman-Yassky et al., 2009*; *Suárez-Fariñas et al., 2013*; *Jabbari et al., 2012*). Overall, our data support a key role for neutrophils in promoting AD itch and inflammation.

## MC903 drives rapid and robust changes in the peripheral and central nervous systems

But how do neutrophils drive AD itch? Itchy stimuli are detected and transduced by specialized subsets of peripheral somatosensory neurons. Thus, to answer this question we first profiled the transcriptional changes in somatosensory neurons in the MC903 model, which were previously unstudied. In general, little is known regarding neuronal changes in chronic itch. Our initial

examination of early hyperinnervation and changes in axon guidance molecules in skin suggested that neurons are indeed affected early on in the MC903 model, before the onset of itch-evoked scratching behaviors. In contrast to the skin, where we saw many early transcriptional changes, we did not see any significant transcriptional changes in the trigeminal ganglia (TG) until five days after the first treatment, and in total only 84 genes were differentially expressed through the eighth day (*Figure 3A–B*). These hits included genes related to excitability of itch sensory neurons, (*Dong and Dong, 2018*; *Usoskin et al., 2015*) neuroinflammatory genes, (*Takeda et al., 2009*) and activity-induced or immediate early genes (*Figure 3A*). Interestingly, we observed enrichment of neuronal markers expressed by one specific subset of somatosensory neurons that are dedicated to itch (*Il31ra, Osmr, Trpa1, Cysltr2*, and *Nppb*), termed 'NP3' neurons (*Dong and Dong, 2018*; *Usoskin et al., 2015*; *Huang et al., 2018*; *Solinski et al., 2019b*). Similar to what has been reported in mouse models of chronic pain, we observed changes in neuroinflammatory (*Bdnf, Nptx1, Nptx2, Nptxr*) and immune genes (*Itk, Cd19, Rag, Tmem173*). However, these transcriptional changes occurred just a few days after itch onset, in contrast to the slow changes in nerve injury and pain models that occur over weeks, indicating that neuropathic changes may occur sooner than previously thought in chronic itch. These changes occurred in tandem with the onset of scratching behaviors (*Figure 1C*), suggesting that the early molecular and cellular changes we observed by this time point may be important for development or maintenance of itch-evoked scratching.

The changes we observed in immune-related genes in the TG were suggestive of infiltration or expansion of immune cell populations, which has been reported in models of nerve injury and chronic pain, but has never been reported in chronic itch. To validate our observations, we used IHC to ask whether CD45$^+$ immune cells increase in the TG. We observed a significant increase in TG immune cell counts at day eight but not day five (*Figure 3C–F*, *Figure 3—figure supplement 1A–D*). Because we observed such dramatic expression changes in the TG on day eight of the model, we postulated that the CNS may also be affected by this time point. Thus, we performed RNA-seq on spinal cord segments that innervate the MC903-treated rostral back skin of mice. To date, only one study has examined changes in the spinal cord during chronic itch (*Shiratori-Hayashi et al., 2015*). The authors showed that upregulation of the STAT3-dependent gene *Lcn2* occurred three weeks after induction of chronic itch and was essential for sustained scratching behaviors. Surprisingly, we saw upregulation of *Lcn2* on day eight of the MC903 model and, additionally, we observed robust induction of immediate early genes (*Fos, Junb, Figure 3G*), suggesting that MC903 itch drives activity-dependent changes in the spinal cord as early as one week after beginning treatment. Together, our findings show that sustained itch and inflammation can drive changes in the PNS and CNS much sooner than previously thought, within days rather than weeks after the onset of scratching. We next set out to explore how loss of neutrophils impacts the molecular changes observed in skin and sensory neurons in the MC903 model, and which of these changes might contribute to neutrophil-dependent itch.

## Neutrophils are required for upregulation of select itch- and atopic-related genes, including the itch-inducing chemokine CXCL10

To ask how neutrophils promote itch in the MC903 model, we examined the transcriptional changes in skin and sensory ganglia isolated from non-itchy neutrophil-depleted animals and from the TSLPR KO mice, which scratched robustly. A number of AD-associated cytokines that were upregulated in control MC903 skin were not upregulated in TSLPR KO and neutrophil-depleted skin. For example, *Il33* upregulation is both neutrophil- and TSLPR-dependent (*Figure 4A*, *Figure 4—figure supplement 1A*). By contrast, upregulation of epithelial-derived cytokines and chemokines *Tslp, Cxcl1, Cxcl2, Cxcl3*, and *Cxcl5* was unaffected by either loss of TSLPR or neutrophil depletion (*Figure 4B*), suggesting these molecules are produced by skin cells even when the MC903-evoked immune response is compromised. Consistent with previous studies, *Il4* upregulation was completely dependent on TSLPR but not neutrophils, establishing a role for TSLP signaling in the Type two immune response. Among the hundreds of MC903-dependent genes we examined, only a handful of genes were uniquely affected by neutrophil depletion. One such gene was *Cxcl10*, a chemokine known to be released by skin epithelial cells, neutrophils, and other myeloid cells (*Hashimoto et al., 2018*; *Ioannidis et al., 2016*; *Kanda et al., 2007*; *Koga et al., 2008*; *Michalec et al., 2002*; *Padovan et al., 2002*; *Tamassia et al., 2007*).*Cxcl10* expression was increased in TSLPR KO but not neutrophil-depleted skin (*Figure 4B*, *Figure 4—figure supplement 1A*). CXCL10 has been

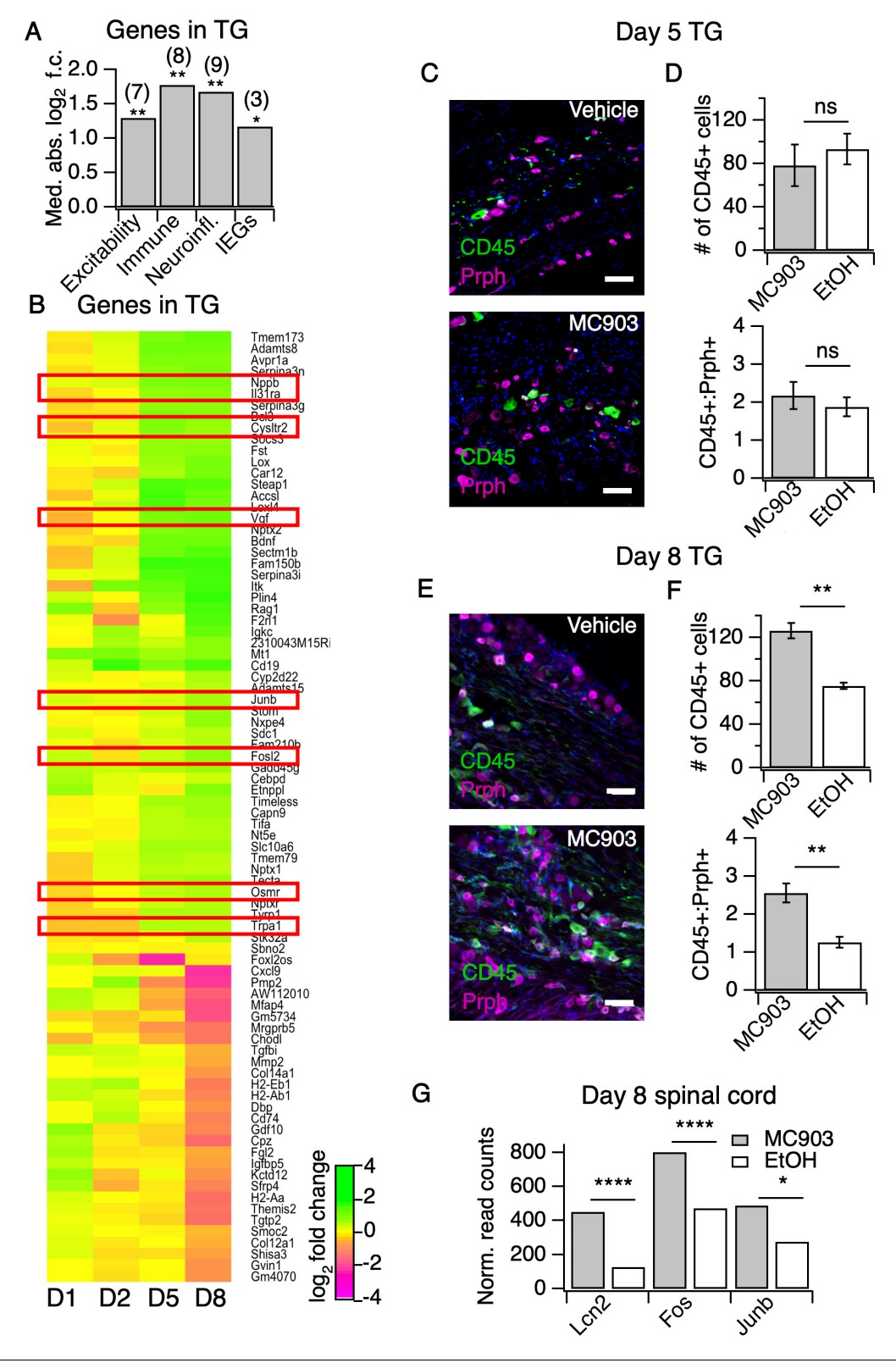

**Figure 3.** The MC903 model induces rapid and robust changes in neuronal tissue. (**A**) Exact permutation test (10,000 iterations, see Materials and methods) for significance of mean absolute log$_2$ fold change in gene expression at Day 8 (MC903 vs. ethanol) of custom-defined groups of genes for indicated categories (see *Figure 3—source data 1*). (**B**) Log$_2$ fold change in gene expression (MC903 vs. ethanol) in mouse trigeminal ganglia (TG) at indicated time points for all genes which were significantly differentially expressed for at least one time point in the MC903 model. *Figure 3 continued on next page*

Figure 3 continued

Green bars = increased expression in MC903 relative to ethanol; magenta = decreased expression. Exact values and corrected *p*-values are reported in *Figure 3—source data 2* and *Source Data 1* Supplemental Data, respectively. (C) Representative composite images showing immune cells (CD45, green), and sensory neurons (Prph, magenta) with DAPI (blue) in sectioned trigeminal ganglia from mice treated with Vehicle or MC903 for five days on the cheek. (D) Quantification of images examining average number of CD45$^+$ cells per section and average ratio of CD45:Peripherin cells per section after five days of treatment (p=0.562 ($t$ = 0.6318, $df$ = 4), 0.542 ($t$ = 0.6660, $df$ = 4); two-tailed unpaired t-tests, n = 33–159 fields of view (images) each of both trigeminal ganglia from three mice per condition treated bilaterally). (E) Representative composite images showing immune cells (CD45, green), and sensory neurons (Peripherin (Prph), magenta) with DAPI (blue) in sectioned trigeminal ganglia from mice treated with Vehicle or MC903 for eight days on the cheek. (F) Quantification of images examining average number of CD45$^+$ cells per section and average ratio of CD45:Peripherin cells per section after eight days of treatment (**p=0.0019 ($t$ = 5.977,$df$ = 5), **p=0.0093 ($t$ = 4.107,$df$ = 4); two-tailed unpaired t-tests; n = 42–172 fields of view (images) each of both trigeminal ganglia from 3 EtOH or 4 MC903 animals treated bilaterally). Scale bar = 100 µm. Images were acquired on a fluorescence microscope using a 10x air objective. Values from bar plots and all TG IHC data are available in *Figure 3—source data 3*. (G) Log$_2$ fold change in gene expression (MC903 vs. ethanol) in mouse spinal cord on day 8 showing selected differentially expressed genes ($p_{adjusted}$ < 0.05). Exact values and corrected *p*-values are reported in *Source Data 1* Supplemental Data.

The online version of this article includes the following source data and figure supplement(s) for figure 3:

**Source data 1.** Values displayed in the bar plot shown in *Figure 3A*.
**Source data 2.** Values displayed in the heat map shown in *Figure 3B*.
**Source data 3.** Quantification of all IHC samples from trigeminal ganglia, and Values displayed in the bar plots shown in *Figure 3D,F*.
**Figure supplement 1.** Method of image quantification for sectioned trigeminal ganglia.

previously shown to drive acute itch in a model of allergic contact dermatitis via CXCR3 signaling in sensory neurons, (*Qu et al., 2015*) and is elevated in skin of AD patients (*Mansouri and Guttman-Yassky, 2015*). Expression of *Cxcl9* and *Cxcl11*, two other CXCR3 ligands that are elevated in AD but have an unknown role in itch, was also decreased in AD skin of neutrophil-depleted mice (*Figure 4B*).

## CXCR3 signaling is necessary for MC903-evoked chronic itch

We hypothesized that neutrophil-dependent upregulation of CXCL10 activates sensory neurons to drive itch behaviors. Consistent with this model, neutrophil depletion attenuated the expression of activity-induced immediate early genes (*Vgf, Junb*) in the TG, suggestive of neutrophil-dependent sensory neuronal activity (*Figure 4C*, *Figure 4—figure supplement 1B*). We found that neutrophils also contributed to other sensory neuronal phenotypes in the model. For example, we observed that expression of *Lcn2*, a marker of neuropathic itch, and activity-induced genes *Fos* and *Junb* were not increased in spinal cord isolated from neutrophil-depleted animals, indicating that neutrophil-dependent scratching behaviors may indeed drive changes in the CNS (*Figure 4D*). We also observed that neutrophil-depleted animals displayed no skin hyperinnervation at day two (*Figure 4E*). This result was surprising because we did not observe significant neutrophil infiltration at this early time point, but these data suggest that low numbers of skin neutrophils are sufficient to mediate these early effects.

To test our model wherein CXCL10 activates CXCR3 to drive neutrophil-dependent itch, we first asked whether this CXCR3 ligand is in fact released in MC903-treated skin. We performed ELISA on cheek skin homogenate and found that CXCL10 protein was increased in MC903-treated skin from uninjected wild-type and TSLPR KO animals, but not in skin from neutrophil-depleted mice (*Figure 4F*). To test whether CXCR3 signaling directly contributes to AD itch, we asked whether acute blockade of CXCR3 using the antagonist AMG 487 (*Qu et al., 2015*) affected scratching behaviors in the MC903 model. We found that the CXCR3 antagonist strongly attenuated scratching behaviors on days five, eight, and twelve (*Figure 4G*), with the greatest effect at day eight. In contrast, CXCR3 blockade did not attenuate scratching behaviors in naive mice injected with the pruritogen chloroquine (*Figure 4G*), demonstrating that CXCR3 signaling contributes to chronic itch but is not required for scratching in response to an acute pruritogen. Thus, we propose that neutrophils promote chronic itch in atopic dermatitis via upregulation of CXCL10 and subsequent activation of CXCR3-dependent itch pathways (*Figure 5*).

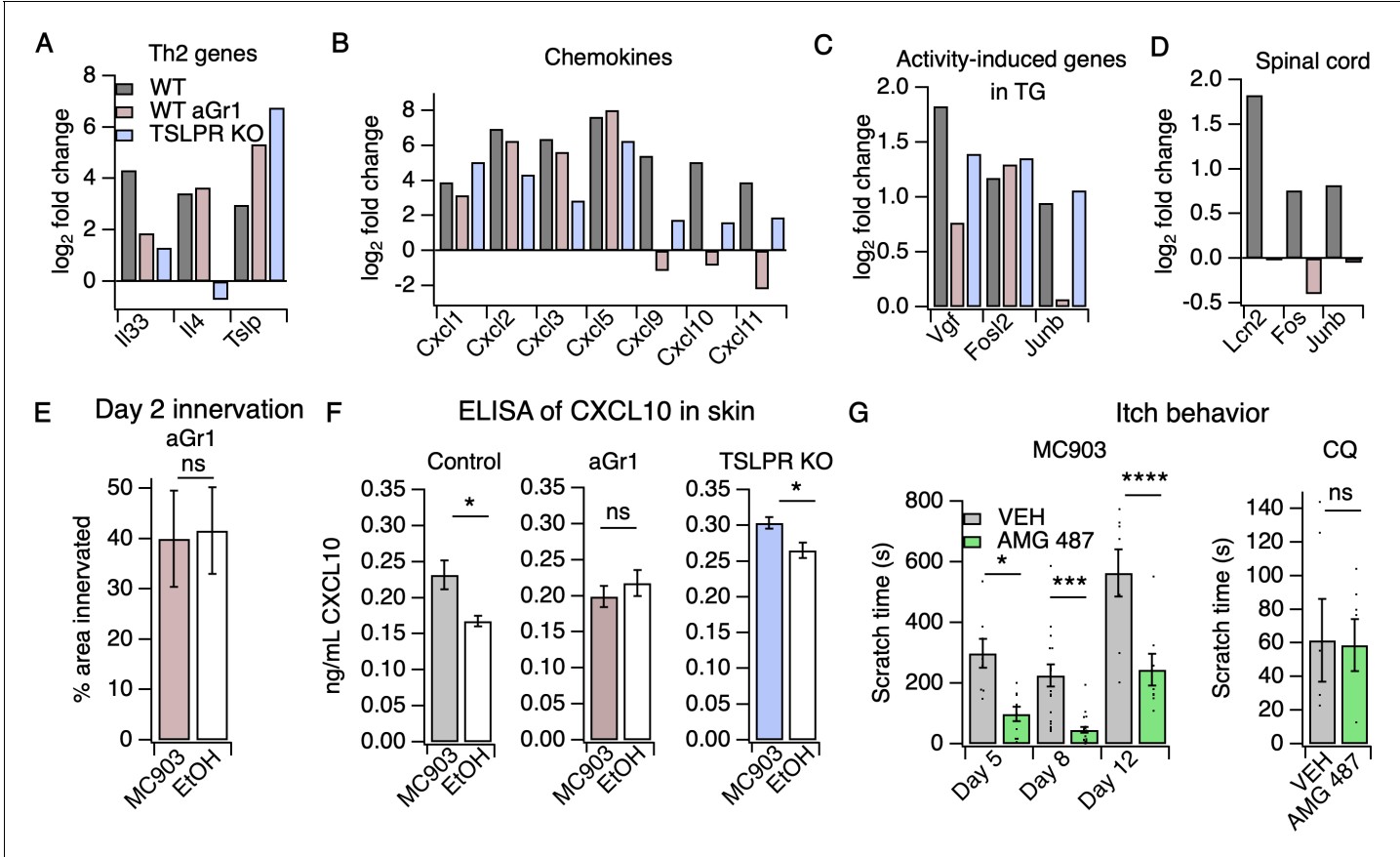

**Figure 4.** Neutrophils are required for induction of the itch-inducing chemokine CXCL10. (A) $Log_2$ fold change (Day 8 MC903 vs. EtOH) of Th2 genes in skin from uninjected wild-type, aGR1-treated, and TSLPR KO animals. (B) $Log_2$ fold change (Day 8 MC903 vs. EtOH) of chemokine genes in skin from uninjected wild-type, aGr1-treated, and TSLPR KO animals. (C) $Log_2$ fold change (Day 8 MC903 vs. EtOH) of activity-induced genes in trigeminal ganglia from uninjected wild-type, aGr1-treated, and TSLPR KO animals. (D) $Log_2$ fold change (Day 8 MC903 vs. EtOH) of *Lcn2* and activity-induced genes in spinal cord from uninjected and aGr1-treated wild-type mice on day 8. For *Figure 4A–D*, exact values and corrected *p*-values are reported in *Source Data 1* Supplemental Data. (E) Quantification of innervation (see Materials and methods) of MC903 and EtOH-treated mouse skin as determined from BTIII staining (p=0.8985; two-tailed t-test ($t = 0.1294$; $df = 18$); n = 9,11 images each from two mice per treatment. Exact values are reported in *Figure 4—source data 1*. (F) CXCL10 levels in skin homogenate as measured by ELISA on day 8 of the MC903 model for uninjected animals (left; *p=0.029 ($t = 2.715$, $df = 7$); two-tailed t-test; n = 4,5 animals), animals which received aGr1 for 8 days (middle; p=0.43 ($t = 0.815$, $df = 11$); two-tailed t-test; n = 6,6 animals), and TSLPR KO animals (right; *p=0.0357 ($t = 2.696$, $df = 6$); two-tailed t-test; n = 4,4 animals. Skin homogenates were isolated on separate days and so uninjected, WT samples were not compared to aGr1-treated samples or to TSLPR KO samples. (G) (Left) Time spent scratching over a thirty minute interval on days 5, 8, and 12 of the MC903 model, one hour after mice were injected with either 3.31 mM of the CXCR3 antagonist AMG 487 or vehicle (20% HPCD in PBS; 50 μL s.c. in rostral back); (two-way ANOVA: ****$p_{treatment}$ <0.0001, F(1,67) = 50.64; Tukey's multiple comparisons: *$p_{day\ 5}$ = 0.0216, n = 8,10 mice; ***$p_{day\ 8}$ = 0.0007, n = 18,21 mice; ****$p_{day\ 12}$ < 0.0001, n = 8,8 mice). (Right) Time spent scratching over a thirty minute interval one hour after mice were injected with either 3.31 mM of the CXCR3 antagonist AMG 487 or vehicle (20% HPCD in PBS; 50 μL s.c. in rostral back), and immediately after mice were injected with 50 mM chloroquine (20 μL i.d., cheek). p=0.92 ($t = 0.0964$, $df = 8$); two-tailed t-test; n = 5,5 mice. Values from bar plots in *Figure 4F–G* are displayed in *Figure 4—source data 2*.

The online version of this article includes the following source data and figure supplement(s) for figure 4:

**Source data 1.** Values displayed in the bar plot shown in *Figure 4E*.
**Source data 2.** Values displayed in the bar plots shown in *Figure 4F–G*.
**Source data 3.** Values displayed in the heat map shown in *Figure 4—figure supplement 1A*.
**Source data 4.** Values displayed in the heat map shown in *Figure 4—figure supplement 1B*.
**Figure supplement 1.** MC903-dependent gene expression changes in aGr1-treated and TSLPR KO animals.

## Discussion

There is great interest in unraveling the neuroimmune interactions that promote acute and chronic itch. Here, we show that neutrophils are essential for the early development of MC903-evoked itch.

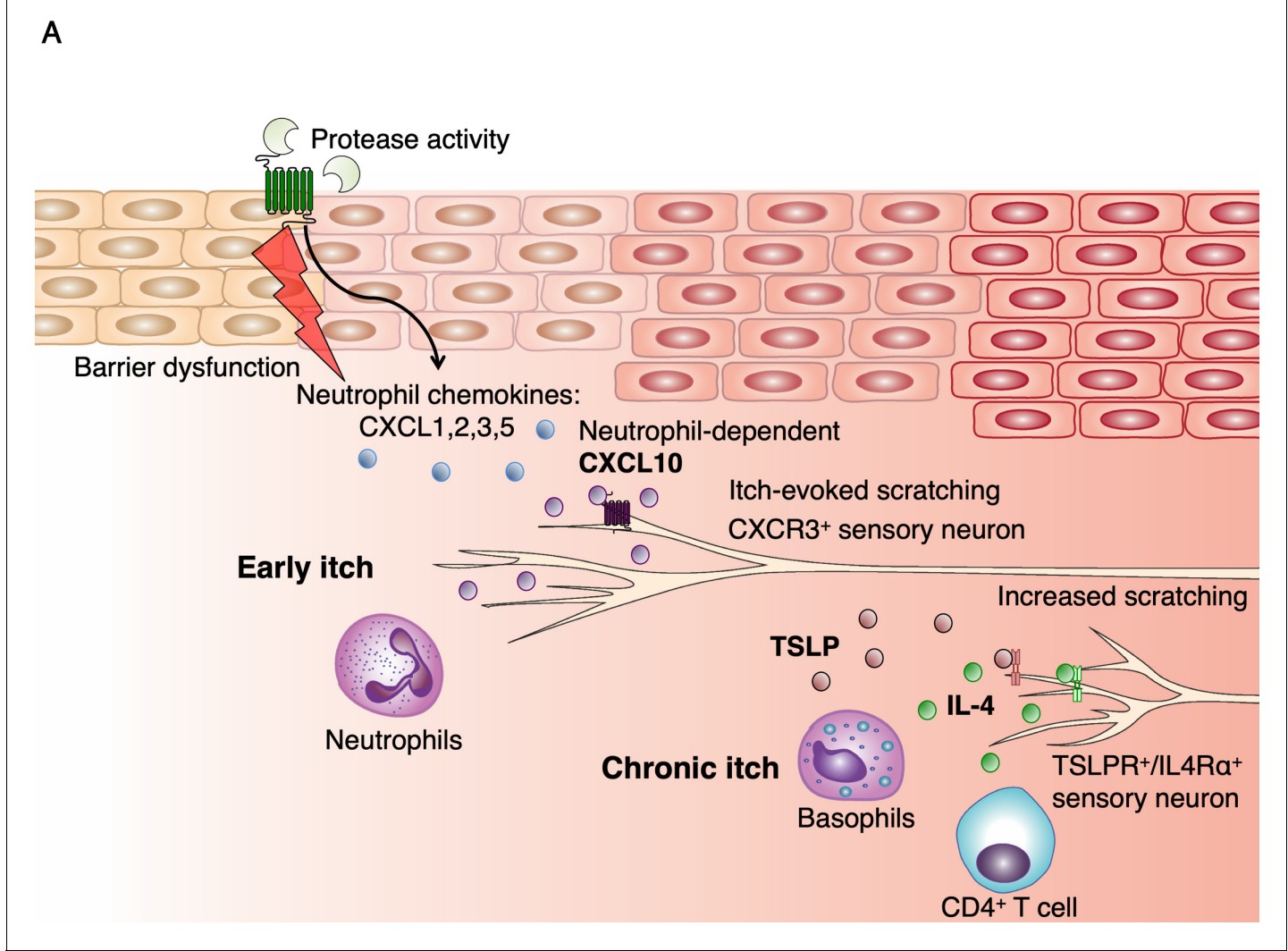

**Figure 5.** Model of early AD pathogenesis. (**A**) AD induction first results in increased protease expression and barrier dysfunction, which drives production of the cytokines TSLP and CXCL1 via PAR2 activation within keratinocytes. CXCL1 can recruit neutrophils via its receptor CXCR2. Neutrophils may evoke itch by multiple pathways, including degranulation and release of proteases and histamine, production of sensitizing lipids such as $PGE_2$ and $LTB_4$, (*Hashimoto et al., 2018*) and induction of CXCL10 expression, which can activate sensory neurons via CXCR3. TSLP activates a number of immune cells to elicit IL-4 production, including basophils, which results in increased IL-4, recruitment of CD4+ T cells, (*Oetjen et al., 2017*) and sensitization of neurons to promote itch later in the model.

We further show that the recruitment of neutrophils to the skin is sufficient to drive itch behaviors within minutes of infiltration. While neutrophils are known to release a variety of pruritogens, their roles in itch and AD were not studied (*Hashimoto et al., 2018*). Only a few studies have even reported the presence of neutrophils in human AD lesions (*Choy et al., 2012*; *Koro et al., 1999*; *Mihm et al., 1976*; *Shalit et al., 1987*). Neutrophils have been implicated in psoriatic inflammation and inflammatory pain, (*Sumida et al., 2014*; *Perkins and Tracey, 2000*; *Guerrero et al., 2008*; *Cunha et al., 2003*; *Finley et al., 2013*; *Carreira et al., 2013*; *Levine et al., 2006*; *Schön et al., 2000*) where they are thought to rapidly respond to tissue injury and inflammation, (*Oyoshi et al., 2012*) but they have not been directly linked to itch.

There is a strong precedence for immune cell-neuronal interactions that drive modality-specific outcomes, such as itch versus pain, under distinct inflammatory conditions. In allergy, mast cells infiltrate the upper dermis and epidermis and release pruritogens to cause itch, (*Solinski et al., 2019b*; *Meixiong et al., 2019*) whereas in tissue injury, mast cell activation can trigger pain hypersensitivity (*Chatterjea and Martinov, 2015*). Likewise, neutrophils are also implicated in both pain and itch.

For example, pyoderma gangrenosum, which causes painful skin ulcerations recruits neutrophils to the deep dermal layers to promote tissue damage and pain (*Hashimoto et al., 2018*). In AD, neutrophils are recruited to the upper dermis and epidermis, (*Choy et al., 2012*; *Shalit et al., 1987*) and we now show that neutrophils trigger itch in AD. Adding to the complex and diverse roles of neutrophils, neutrophils recruited to subcutaneous sites during invasive streptococcal infection alleviate pain by clearing the tissue of bacteria (*Pinho-Ribeiro et al., 2018*). Several potential mechanisms may explain these diverse effects of neutrophils. First, the location of the inflammatory insult could promote preferential engagement of pain versus itch nerve fibers (*Hashimoto et al., 2018*). This is supported by observations that neutrophil-derived reactive oxygen species and leukotrienes can promote either itch or pain under different inflammatory conditions (*Salvemini et al., 2011*; *Bautista et al., 2006*; *Liu and Ji, 2012*; *Caceres et al., 2009*). Second, it has been proposed that there are distinct functional subsets of neutrophils that release modality-specific inflammatory mediators (*Wang, 2018*). Third, the disease-specific inflammatory milieu may induce neutrophils to specifically secrete mediators of either itch or pain. Indeed, all three of these mechanisms have been proposed to underlie the diverse functions of microglia and macrophages in homeostasis, tissue repair, injury, and neurodegenerative disease (*Hammond et al., 2018*). It will be of great interest to the field to decipher the distinct mechanisms by which neutrophils and other immune cells interact with the nervous system to drive pain and itch.

In addition to neutrophils, TSLP signaling and the Type two immune response plays an important role in the development of itch in the second week of the MC903 model. Dendritic cells, mast cells, basophils, and CD4[+] T cells are all major effectors of the TSLP inflammatory pathway in the skin. We propose that neutrophils play an early role in triggering itch and also contribute to chronic itch in parallel with the TSLP-Type two response. While we have ruled out an early role for TSLP signaling and basophils and CD4[+] T cells in early itch, other cell types such as mast cells, which have recently been linked directly to chronic itch, (*Solinski et al., 2019b*; *Meixiong et al., 2019*) and dendritic cells may be playing an important role in setting the stage for itch and inflammation prior to infiltration of neutrophils.

Given the large magnitude of the itch deficit in the neutrophil-depleted mice, we were surprised to find fewer expression differences in MC903-dependent, AD-associated genes between neutrophil depleted and non-depleted mice than were observed between WT and TSLPR KO mice. One of the few exceptions were the Th1-associated genes *Cxcl9/10/11* (*Ewald et al., 2017*; *Brunner et al., 2017*). We found that induction of these genes and of CXCL10 protein was completely dependent on neutrophils. While our results do not identify the particular cell type(s) responsible for neutrophil-dependent CXCL10 production, a number of cell types present in skin have been shown to produce CXCL10, including epithelial keratinocytes, myeloid cells, and sensory neurons (*Hashimoto et al., 2018*; *Ioannidis et al., 2016*; *Kanda et al., 2007*; *Koga et al., 2008*; *Michalec et al., 2002*; *Padovan et al., 2002*; *Tamassia et al., 2007*). In support of a role for neutrophils in promoting chronic itch, we observed striking differences in neutrophil-dependent gene expression in the spinal cord, where expression of activity-induced genes and the chronic itch gene *Lcn2* were markedly attenuated by loss of neutrophils. Moreover, we also demonstrate that depletion of neutrophils in the second week of the MC903 model can attenuate chronic itch-evoked scratching. In examining previous characterizations of both human and mouse models of AD and related chronic itch disorders, several studies report that neutrophils and/or neutrophil chemokines are indeed present in chronic lesions (*Ewald et al., 2017*; *Choy et al., 2012*; *Guttman-Yassky et al., 2009*; *Suárez-Fariñas et al., 2013*; *Jabbari et al., 2012*; *Nattkemper et al., 2018*; *Li et al., 2017*; *Saunders et al., 2016*; *Andersson, 2015*; *Liu et al., 2019*; *Malik et al., 2017*). Our observations newly implicate neutrophils in setting the stage for the acute-to-chronic itch transition by triggering molecular changes necessary to develop a chronic, itchy lesion and also contributing to persistent itch.

Additionally, we demonstrate a novel role of CXCR3 signaling in MC903-induced itch. The CXCR3 ligand CXCL10 contributes to mouse models of acute and allergic itch (*Qu et al., 2015*; *Qu et al., 2017*; *Jing et al., 2018*); however, its role in chronic itch was previously unknown. We speculate that the residual itch behaviors after administration of the CXCR3 antagonist could be due to TSLPR-dependent IL-4 signaling, as TSLPR-deficient mice display reduced itch behaviors by the second week of the model, or due to some other aspect of neutrophil signaling, such as release of proteases, leukotrienes, prostaglandins, or reactive oxygen species, all of which can directly trigger

itch via activation of somatosensory neurons (*Hashimoto et al., 2018*). Our observations are in alignment with a recent study showing that dupilumab, a new AD drug that blocks IL4Rα, a major downstream effector of the TSLP signaling pathway, does not significantly reduce CXCL10 protein levels in human AD lesions (*Hamilton et al., 2014*). Taken together, these findings suggest that the TSLP/IL-4 and neutrophil/CXCL10 pathways are not highly interdependent, and supports our findings that *Il4* transcript is robustly upregulated in the absence of neutrophils. Additionally, targeting IL4Rα signaling has been successful in treating itch and inflammation in some, but not all, AD patients (*Simpson et al., 2016*). We propose that biologics or compounds targeting neutrophils and/or the CXCR3 pathway may be useful for AD that is incompletely cleared by dupilumab monotherapy. Drugs targeting neutrophils are currently in clinical trials for the treatment of psoriasis, asthma, and other inflammatory disorders. For example, MDX-1100, a biologic that targets CXCL10, has already shown efficacy for treatment of rheumatoid arthritis in phase II clinical trials (*Yellin et al., 2012*). While rheumatoid arthritis and AD have distinct etiologies, (*Scott et al., 2010*) our body of work indicates that CXCL10 or CXCR3 may be promising targets for treating chronic itch. Our findings may also be applicable to other itch disorders where neutrophil chemoattractants and/or CXCL10 are also elevated, such as psoriasis and allergic contact dermatitis. Overall, our data suggest that neutrophils incite itch and inflammation in early AD through several mechanisms, including: 1) directly triggering itch upon infiltration into the skin, as shown by acute injection of CXCL1, and, 2) indirectly triggering itch by altering expression of endogenous pruritogens (e.g. induction of *Cxcl10* expression; *Hashimoto et al., 2018*; *Ioannidis et al., 2016*; *Kanda et al., 2007*; *Koga et al., 2008*; *Michalec et al., 2002*; *Padovan et al., 2002*; *Tamassia et al., 2007*). Together, these direct and indirect mechanisms for neutrophil-dependent itch may explain why neutrophils have a dramatic effect on scratching behaviors on not only days eight and twelve but also day five of the model, when neutrophils are recruited in large numbers, but CXCR3 ligands are not as robustly induced.

More generally, our study provides a framework for understanding how and when human chronic itch disease genes contribute to the distinct stages of AD pathogenesis. Our analysis of MC903-evoked transcriptional changes suggests we may be able to extend findings in the model not only to atopic dermatitis, but also to related disorders, including specific genetic forms of atopy. For example, we provide evidence that MC903 treatment may also model the filaggrin loss-of-function mutations, which are a key inciting factor in human heritable atopic disease (*Palmer et al., 2006*; *Sandilands et al., 2007*). There are many rich datasets looking at mature patient lesions and datasets for mature lesions in other mouse models of chronic itch (*Ewald et al., 2017*; *Choy et al., 2012*; *Guttman-Yassky et al., 2009*; *Jabbari et al., 2012*; *Nattkemper et al., 2018*; *Oetjen et al., 2017*; *Liu et al., 2019*; *Liu et al., 2016*). Our study adds a temporal frame of reference to these existing datasets and sets the stage for probing the function of AD disease genes in greater detail. Furthermore, we have mapped the time course of gene expression changes in primary sensory ganglia and spinal cord during chronic itch development. We show that the MC903 model recapitulates several hallmarks of neuropathic disease on a time course much shorter than has been reported for chronic itch, or chronic pain. Nervous system tissues are extremely difficult to obtain from human AD patients, and thus little is known regarding the neuronal changes in chronic itch disorders in both mouse models and human patients. Our findings can now be compared to existing and future datasets examining neuronal changes in chronic pain, diabetic neuropathy, shingles, neuropathic itch, psoriasis, and other inflammatory disorders where neuronal changes are poorly understood but may contribute to disease progression. The early changes we see in skin innervation, sensory ganglia, and spinal cord dovetail with recent studies examining neuroimmune interactions in other inflammatory conditions, (*Pinho-Ribeiro et al., 2018*; *Baral et al., 2018*; *Pinho-Ribeiro et al., 2017*; *Blake et al., 2018*) which all implicate early involvement of sensory neurons in the pathogenesis of inflammatory diseases.

## Materials and methods

### Key resources table

| Reagent type (species) or resource | Designation | Source or reference | Identifiers | Additional information |
|---|---|---|---|---|

*Continued on next page*

*Continued*

| Reagent type (species) or resource | Designation | Source or reference | Identifiers | Additional information |
|---|---|---|---|---|
| Strain, strain background (*Mus musculus*) | C57BL/6; WT; wild-type | The Jackson Laboratory | Jackson Stock #: 000664; RRID:IMSR_JAX:000664 | |
| Strain, strain background (*Mus musculus*) | C57BL/6; WT; wild-type | Charles River Laboratories | RRID:IMSR_CRL:27; Charles River strain code #: 027; | |
| Strain, strain background (*Mus musculus*) | Crlf2tm1Jni; TSLPR KO | PMID: 14993294 | RRID:MGI:3039553; MGI Cat# 3039553 | Obtained from the laboratory of Steven F. Ziegler (Ben Aroya Research Institute) |
| Antibody | Purified anti-mouse Ly-6G/Gr-1 antibody. Low endotoxin, no azide, in PBS; anti-GR1 (RB6-8C5); aGr1 | UCSF Core | UCSF Core Cat# AM051 | Obtained from the laboratory of Daniel Portnoy (UC Berkeley) |
| Antibody | LEAF Purified anti-mouse Ly-6G/Ly-6C (Gr-1); antibody; RB6-8C5; aGr1 | Biolegend | RRID:AB_313379; BioLegend Cat# 108414 | |
| Antibody | Anti-β-tubulin III (Rabbit polyclonal; 1:1000) | Abcam | RRID:AB_444319; Cat # ab18207 | |
| Antibody | Anti-CGRP (Rabbit polyclonal; 1:1000) | Immunostar | RRID:AB_572217; Cat # 24112 | |
| Antibody | Anti-Peripherin (Chicken polyclonal; 1:1000) | Abcam | RRID:AB_777207; Cat # ab39374 | |
| Antibody | Goat Anti-Mouse IgG H and L Alexa Fluor 488 (Goat polyclonal; 1:1000) | Abcam | RRID:AB_2688012; Cat # ab150117 | |
| Antibody | Goat anti-Chicken IgY (H+L) Secondary Antibody, Alexa Fluor 488 (Goat polyclonal; 1:1000) | Thermo Fisher Scientific | RRID:AB_2534096; Cat # A-11039 | |
| Antibody | Goat Anti-Chicken IgG (H+L) Secondary Antibody, Alexa Fluor 594 (Goat polyclonal; 1:1000) | Thermo Fisher Scientific | RRID:AB_2534099; Cat # A11042 | |
| Antibody | Goat anti-Rabbit IgG (H+L) Secondary Antibody, Alexa Fluor 594 (Goat polyclonal; 1:1000) | Invitrogen | RRID:AB_2556545; Cat # R37117 | |
| Commercial assay or kit | Promocell Keratinocyte Growth Medium 2 | Promocell | Cat # C-20011 | |
| Cell line (human) | Normal Human Epidermal Keratinocytes (NHEK), single juvenile donor, cryopreserved | Promocell | Cat # C-12001 | |
| Other | Liberase TM Research Grade; Liberase | Roche | Cat # 5401119001 | |

*Continued on next page*

*Continued*

| Reagent type (species) or resource | Designation | Source or reference | Identifiers | Additional information |
|---|---|---|---|---|
| Other | Dnase I from bovine pancreas | Sigma | Cat # 11284932001 | |
| Other | Ambionª DNase I (RNase-free); DNAse | Ambion | Cat # AM2222 | |
| Peptide, recombinant protein | SLIGRL-NH2; SLIGRL | Tocris | Cas 171436-38-7; Cat #1468 | |
| Commercial assay or kit | Qiagen RNeasy mini kit | Qiagen | Cat # 74104 | |
| Commercial assay or kit | RNAzol RT | Sigma-Aldrich | Cat # R4533-50ML | |
| Chemical compound, drug | (2-Hydroxypropyl)-β-cyclodextrin; HPCD | Sigma-Aldrich | Cas 128446-35-5; Cat # H107 | |
| Chemical compound, drug | Methyl alcohol; Methanol; MeOH | Sigma-Aldrich | Cas 67-56-1; Cat # 34860 | |
| Chemical compound, drug | Ethanol, Absolute (200 Proof), Molecular Biology Grade, Fisher BioReagents; Absolute Ethanol, Molecular-Biology grade; Ethanol; EtOH | Fischer Scientific | Cas 64-17-5; Cat # BP2818100 | |
| Chemical compound, drug | MC903; Calcipotriol | Tocris | Cas 112965-21-6; Cat # 2700 | |
| Chemical compound, drug | (±)-AMG 487; AMG | Tocris | Cas 947536-03-0; Cat # 4487 | |
| Chemical compound, drug | Chloroquine diphosphate; Chloroquine | Sigma-Aldrich | CAS 50-63-5; Cat # C6628 | |
| Chemical compound, drug | Dimethyl sulfoxide; DMSO | Sigma-Aldrich | Cat # 8418–100 mL | |
| Chemical compound, drug | Formaldehyde, 16%, methanol free, Ultra Pure; Paraformaldehyde; PFA | Polysciences, Inc. | Cat # 18814–10 | |
| Chemical compound, drug | Tissue Tek Optimal cutting temperature compound (OCT) | Sakura Finetek USA | Cat # 4583 | |
| Chemical compound, drug | Triton X-100 solution; Triton X-100 | BioUltra | CAS 9002-93-1; Cat # 93443 | |
| Chemical compound, drug | Phosphate-buffered saline (PBS), pH 7.4; PBS | Gibco | Cat # 10010023 | |
| Chemical compound, drug | Benzyl benzoate | Sigma-Aldrich | CAS 120-51-4; Cat # B6630 | |
| Chemical compound, drug | Benzyl alcohol | Sigma-Aldrich | CAS 100-51-6; Cat # 305197 | |
| Chemical compound, drug | Sucrose | Sigma-Aldrich | CAS 57-50-1; Cat # S0389 | |
| Chemical compound, drug | LIVE/DEAD Fixable Aqua Dead Cell Stain Kit, for 405 nm excitation; Aqua | Thermo Fisher Scientific | Cat # L34957 | |
| Chemical compound, drug | Isoflurane | Piramal | CAS 26675-46-7 | |

*Continued on next page*

Continued

| Reagent type (species) or resource | Designation | Source or reference | Identifiers | Additional information |
|---|---|---|---|---|
| Chemical compound, drug | 4',6-Diamidino-2-Phenylindole, Dihydrochloride; DAPI | ThermoFisher Scientific | CAS 28718-90-3; Cat # 1306 | |
| Chemical compound, drug | 4',6-Diamidino-2-Phenylindole, Dihydrochloride; DAPI LIVE/DEAD | Invitrogen | Cat # L34961 | |
| Chemical compound, drug | Fluoromount-G | ThermoFisher Scientific | Cat # 00-4958-02 | |
| Antibody | Goat Anti-Mouse IgG - H and L - Fab Fragment Polyclonal Antibody, Unconjugated, Abcam; F(ab) anti-mouse IgG ( Goat polyclonal; 1:200) | Abcam | RRID:AB_955960; Cat # ab6668 | |
| Antibody | Anti-Mouse CD45.2 Purified 100 ug antibody, Thermo Fisher Scientific; Mouse anti-CD45.2 (Mouse monoclonal; 1:1000) | eBioscience | RRID:AB_467261; Cat # 14-0454-82 | |
| Antibody | Purified anti-mouse CD16/32 antibody. Low endotoxin, no azide, in PBS; Rat anti-Mouse CD16/32 (2.4G2) (Rat monoclonal; 1:1000) | UCSF Core | UCSF Core Cat# AM004 | |
| Commercial assay or kit | DuoSet ELISA Ancillary Reagent Kit 2 | R and D Systems | Cat # DY008 | |
| Commercial assay or kit | Mouse CXCl10 DuoSet ELISA | R and D Systems | Cat # DY466 | |
| Commercial assay or kit | Pierce BCA Protein Assay Kit - Reducing Agent Compatible | Thermo Fisher Scientific | Cat # 23250 | |
| Chemical compound, drug | 2-Amino-2-(hydroxymethyl) −1,3-propanediol; Trizma base, TRIS, TRIS base | Sigma-Aldrich | Cas 77-86-1; Cat # T4661 | |
| Chemical compound, drug | Ethylene glycol-bis (2-aminoethylether)-N,N,N',N'-tetraacetic acid; EGTA | Sigma-Aldrich | Cas 67-42-5; Cat # E3889 | |
| Chemical compound, drug | Ethylenedinitrilo) tetraacetic acid; EDTA | Sigma-Aldrich | Cas 60-00-4; Cat # E9884 | |
| Commercial assay or kit | PhosSTOP inhibitor | Roche | Cat # 4906845001 | |
| Chemical compound, drug | Sodium deoxycholate,≥97% (titration); Sodium deoxycholate | Sigma-Aldrich | Cas 302-95-4; Cat # D6750 | |

*Continued*

| Reagent type (species) or resource | Designation | Source or reference | Identifiers | Additional information |
|---|---|---|---|---|
| Chemical compound, drug | Phenylmethyl sulfonyl fluoride; PMSF | Sigma-Aldrich | Cas 329-98-6; Cat # 10837091001 | |
| Chemical compound, drug | 1-Fluoro-2,4, -dinitrobenzene; DNFB | Sigma | Cas 70-34-8; Cat # D1529 | |
| Commercial assay or kit | cOmplete protease inhibitor cocktail | Roche | Cat # 11697498001 | |
| Other | Advanced RPMI Medium 1640; RPMI | Gibco | Cat # 12633012 | |
| Other | Fetal Bovine Serum; FBS; FCS | HyClone | Cat # 30396.03 | |
| Other | sodium pyruvate 100 mM | Gibco | Cat # 11360070 | |
| Other | N-2-hydroxyethy lpiperazine-N-2- ethane sulfonic acid; HEPES 1M | Gibco | Cat # 15630080 | |
| Other | L-Glutamine 200 mM | Gibco | Cat # 25030081 | |
| Other | Penicillin-Streptomycin (10,000 U/mL; Pen-Strep | Gibco | Cat # 15140122 | |
| Other | Collagenase VIII | Sigma-Aldrich | Cat # C2139-500MG | |
| Commercial assay or kit | Invitrogen CountBright Absolute Counting Beads, for flow cytometry; Counting Beads | Invitrogen | Cat # C36950 | |
| Antibody | CD45 Monoclonal Antibody (30-F11), APC-eFluor 780, eBioscience(TM), Thermo Fisher Scientific; CD45-APC/eFluor 780 (30-F11) (Rat monoclonal; 1:200) | eBioscience | RRID:AB_1548781; Cat # 47-0451-82 | |
| Antibody | CD11b Monoclonal Antibody (M1/70), PE-Cyanine7, eBioscience(TM), Thermo Fisher Scientific; CD11b-PE/Cy7 (M1/70) (Rat monoclonal; 1:200) | BD Biosciences | RRID:AB_469588; Cat # 25-0112-82 | |
| Antibody | PE-Cyanine7 Anti-Human/Mouse CD45R (B220) (RA3-6B2) Antibody, Tonbo Biosciences; B220-PE/Cy7 ( RA3-6B2) (Rat monoclonal; 1:200) | Tonbo Biosciences | RRID:AB_2621849; Cat # 60–0452 | |
| Antibody | CD11c Monoclonal Antibody (N418), PE-Cyanine7, eBioscience(TM), Thermo Fisher Scientific; CD11c-PE/Cy7 (N418) (Armenian Hamster monoclonal; 1:200) | eBioscience | RRID:AB_469590; Cat # 25-0114-82 | |

*Continued on next page*

Continued

| Reagent type (species) or resource | Designation | Source or reference | Identifiers | Additional information |
|---|---|---|---|---|
| Antibody | CD3e Monoclonal Antibody (145–2 C11), FITC, eBioscience(TM), Thermo Fisher Scientific; CD3-FITC (145–2 C11) (Armenian Hamster monoclonal; 1:200) | eBioscience | RRID:AB_464882; Cat # 11-0031-82 | |
| Antibody | Brilliant Violet 785 anti-mouse CD8a antibody, BioLegend; CD8-BV785 (53–6.7) (Rat monoclonal; 1:200) | Biolegend | RRID:AB_1121880; Cat # 100749 | |
| Antibody | Rat Anti-CD4 Monoclonal Antibody, Phycoerythrin Conjugated, Clone GK1.5, BD Biosciences; CD4-PE (GK1.5) (Rat monoclonal; 1:200) | BD Biosciences | RRID:AB_395014; Cat # 553730 | |
| Antibody | Alexa Fluor 647 anti-mouse TCR γ/δ Antibody; gdTCR-AF647 (GL3) (Armenian Hamster monoclonal; 1:200) | Biolegend | RRID:AB_313826; Cat # 118133 | |
| Antibody | CD117 (c-Kit) Monoclonal Antibody (2B8), Biotin; c-Kit-Biotin (ACK2) (Rat monoclonal; 1:200) | eBioscience | RRID:AB_466569; Cat # 13-1171-82 | |
| Antibody | FceR1 alpha Monoclonal Antibody (MAR-1), PE, eBioscience; FceRI-PE (MAR-1) (Armenian Hamster monoclonal; 1:200) | eBioscience | RRID:AB_466028; Cat # 12-5898-82 | |
| Antibody | CD49b (Integrin alpha 2) Monoclonal Antibody (DX5), PE-Cyanine7, eBioscience; CD49b-PE/Cy7 (DX5) (Rat monoclonal; 1:200) | eBioscience | RRID:AB_469667; Cat # 25-5971-82 | |
| Antibody | Anti-Siglec-F-APC, mouse (clone: REA798); SiglecF-APC; (Rat monoclonal; 1:200) | Miltenyi Biotech | RRID:AB_2653441; Cat # 130-112-333 | |
| Other | Streptavidin FITC; SA-FITC | eBioscience | RRID:AB_11431787; Cat # 11-4317-87 | |
| Antibody | Ly-6C Monoclonal Antibody (HK1.4), PerCP-Cyanine5.5, eBioscience; Ly6C-PerCP/Cy5.5 (HK1.4) (Rat monoclonal; 1:200) | eBioscience | RRID:AB_1518762; Cat # 45-5932-82 | |

TSLPR KO mice were kindly provided by Dr. Steven Ziegler (*Crlf2*<sup>tm1Jni</sup>; *Carpino et al., 2004*) and backcrossed onto C57BL/6. All experiments were performed under the policies and recommendations of the International Association for the Study of Pain and approved by the University of California Berkeley Animal Care and Use Committee. Where appropriate, genotypes were assessed using standard PCR.

## MC903 model of atopic dermatitis

MC903 (Calcipotriol; R and D Systems) was applied to the shaved mouse cheek (20 µl of 0.2 mM in ethanol) or rostral back (40 µl of 0.2 mM in ethanol) once per day for 1–12 days using a pipette. 100% ethanol was used. All MC903 studies were performed on 8–12 week old age-matched mice. Behavior, RNA-seq, flow cytometry, and immunohistochemistry were performed on days 1, 2, 3, 5, eight and/or 12. For AMG 487 experiments in the MC903 model, 50 µL 3.31 mM AMG 487 (Tocris) or 20% HPCD-PBS vehicle was injected subcutaneously one hour prior to recording behavior (*Qu et al., 2015*). Spontaneous scratching was manually scored for the first 30 min of observation. Both bout number and length were recorded. Behavioral scoring was performed while blind to experimental condition and mouse genotype.

## MC903 RNA isolation and sequencing

On days 1 (six hours post-treatment), 2, 5, or eight post-treatment, mice treated with MC903 and vehicle were euthanized via isoflurane and cervical dislocation. Cheek skin was removed, flash-frozen in liquid nitrogen, and cryo-homogenized with a mortar and pestle. Ipsilateral trigeminal ganglia were dissected and both skin and trigeminal ganglia were homogenized for three minutes (skin) or one minute (TG) in 1 mL RNAzol RT (Sigma-Aldrich). Thoracic spinal cord was dissected from mice treated with 40 µL MC903 or ethanol on the shaved rostral back skin and homogenized for one minute in 1 mL RNAzol. Large RNA was extracted using RNAzol RT per manufacturer's instructions. RNA pellets were DNase treated (Ambion), resuspended in 50 µL DEPC-treated water, and subjected to poly(A) selection and RNA-seq library preparation (Apollo 324) at the Functional Genomics Laboratory (UC Berkeley). Single-end read sequencing (length = 50 bp) was performed by the QB3 Vincent G. Coates Genomic Sequencing Laboratory (UC Berkeley) on an Illumina HiSeq4000. See *Supplementary file 1* for number of mice per experimental condition and number of mapped reads per sample. Data are available at Gene Expression Omnibus under GSE132173.

## MC903 RNA sequencing analysis

Reads were mapped to the mm10 mouse genome using Bowtie2 and Tophat, and reads were assigned to transcripts using htseq-count (*Langmead et al., 2009*; *Langmead and Salzberg, 2012*). For a given time point, replicate measurements for each gene from treated and control mice were used as input for DESeq (R) and genes with $p_{adjusted} < 0.05$ (for skin and spinal cord) or $p_{adjusted} < 0.1$ (for trigeminal ganglia) for at least one time point were retained for analysis (*Anders and Huber, 2012*; *Anders et al., 2013*). For the skin dataset, we collated a set of AD-related immune cell markers, cytokines, atopic dermatitis disease genes, neurite outgrown/axonal guidance genes, and locally expressed neuronal transcripts, and from this list visualized genes that were significantly differentially expressed for at least one time point. For the trigeminal ganglia dataset, we plotted all genes that were significantly differentially expressed for at least one time point. Genes from these lists were plotted with hierarchical clustering using heatmap2 (R) (*Hill, 2019*).

## Custom gene groups

Genes were clustered into functional groups and significance was evaluated using a permutation test. Briefly, we first tabulated the absolute value of the $\log_2$ fold change of gene expression (between MC903 and EtOH) of each gene in a given group of $n$ genes in turn, and then we calculated the median of these fold change values, $z_{true}$. We then drew $n$ random genes from the set of all genes detected in the samples and computed the median $\log_2$ fold change as above using this null set, $z_{null}$. Repeating the latter 10,000 times established a null distribution of median $\log_2$ fold change values; we took the proportion of resampled gene groups that exhibited ($z_{true} \geq z_{null}$) as an empirical *p*-value reporting the significance of changes in gene expression for a given group of $n$ genes.

## Flow cytometry

Skin samples were collected from the cheek of mice at the indicated time points with a 4- or 6 mm biopsy punch into cold RPMI 1640 medium (RPMI; Gibco) and minced into smaller pieces with surgical scissors. When ear skin was collected, whole ears were dissected postmortem into cold RPMI and finely minced with scissors. For isolation of immune cells, skin samples were digested for 1 hr at 37 ˚C using 1 U/mL Liberase TM (Roche) and 5 µg/mL DNAse I (Sigma). At the end of the digestion, samples were washed in FACS buffer (PBS with 0.5% FCS and 2 mM EDTA) and filtered through a 70 or 100 µm strainer (Falcon). Cells were stained with LIVE/DEAD fixable stain Aqua in PBS (Invitrogen), then blocked with anti-CD16/32 (UCSF Core) and stained with the following fluorophore-conjugated antibodies (all from eBiosciences unless stated otherwise) in FACS buffer: cKit-Biotin (clone ACK2; secondary stain with SA-FITC), CD11b-violet fluor 450 (Tonbo; clone M1/70), Ly6C-PerCP/ Cy5.5 (clone HK1.4), CD49b-PE/Cy7 (clone DX5), CD45.2-APC/Cy7 (clone 104), FceRI-PE (MAR-1), Ly6G-AF700 (clone 1A8). 10 µL of counting beads (Invitrogen) were added after the last wash to measure absolute cell counts. For measurement of CD4$^+$ T cells, 6 mm skin biopsy punch samples were digested for 30 min at 37 ˚C using Collagenase VIII (Sigma). At the end of the digestion, cells were washed in RPMI buffer (RPMI with: 5% FCS, 1% penicillin-streptomycin, 2 mM L-glutamine, 10 mM HEPES buffer, 1 mM sodium pyruvate). Cells were blocked with anti-CD16/32 (UCSF Core) and stained with the following fluorophore-conjugated antibodies in FACS buffer (PBS with 5% FCS and 2 mM EDTA): CD45-APC-eFluor780 (clone 30-F11; eBiosciences), CD11b-PE/Cy7 (clone M1/70; BD Biosciences), B220-PE/Cy7 (clone RA3-6B2; Tonbo Biosciences), CD11c-PE/Cy7 (clone N418; eBiosciences), CD3-FITC (clone 145–2 C11; eBiosciences), CD8-BV785 (clone 53–6.7; Biolegend), CD4-PE (clone GK1.5; BD Biosciences), gdTCR-AF647 (clone GL3; Biolegend). 10 µL of counting beads (Invitrogen) were added after the last wash to measure absolute cell counts, and samples were resuspended in DAPI LIVE/DEAD (Invitrogen). Blood samples were collected from saphenous vein or from terminal bleed following decapitation. Red blood cells were lysed using ACK lysis buffer (Gibco), and samples were washed with FACS buffer (PBS with 0.5% FCS and 2 mM EDTA), and blocked with anti-CD16/32. Cells were stained with Ly6G-PE (1A8; BD Biosciences), CD11b-violet fluor 450 (M1/ 70, Tonbo), Ly6C-PerCP/Cy5.5 (HK1.4, Biolegend), and aGr1-APC/Cy7 (RB6-8C5, eBiosciences). For all experiments, single cell suspensions were analyzed on an LSR II or LSR Fortessa (BD Biosciences), and data were analyzed using FlowJo (TreeStar, v.9.9.3) software.

## Human keratinocyte RNA sequencing

Normal human epidermal keratinocytes from juvenile skin (PromoCell #C-12001) were cultured in PromoCell Keratinocyte Growth Medium two and passaged fewer than five times. Cells were treated for three hours at room temperature with 100 µM SLIGRL or vehicle (Ringer's + 0.1% DMSO). Total RNA was extracted by column purification (Qiagen RNeasy Mini Kit). RNA was sent to the Vincent J. Coates Sequencing Laboratory at UC Berkeley for standard library preparation and sequenced on an Illumina HiSeq2500 or 4000. Sequences were trimmed (Trimmomatic), mapped (hg19, TopHat) and assigned to transcripts using htseq-count. Differential gene expression was assessed using R (edgeR) (*Hill, 2019*). Data are available at Gene Expression Omnibus under GSE132174.

## IHC of whole-mount skin

Staining was performed as previously described (*Hill et al., 2018*; *Marshall et al., 2016*). Briefly, 8 week old mice were euthanized and the cheek skin was shaved. The removed skin was fixed overnight in 4% PFA, then washed in PBS (3X for 10 min each). Dermal fat was scraped away with a scalpel and skin was washed in PBST (0.3% Triton X-100; 3X for two hours each) then incubated in 1:500 primary antibody (Rabbit anti beta-Tubulin II; Abcam #ab18207 or Rabbit anti-CGRP; Immunostar #24112) in blocking buffer (PBST with 5% goat serum and 20% DMSO) for 6 days at 4˚C. Skin was washed as before and incubated in 1:500 secondary antibody (Goat anti-Rabbit Alexa 594; Invitrogen #R37117) in blocking buffer for 3 days at 4˚C. Skin was washed in PBST, serially dried in methanol: PBS solutions, incubated overnight in 100% methanol, and finally cleared with a 1:2 solution of benzyl alcohol: benzyl benzoate (BABB; Sigma) before mounting between No. 1.5 coverglass. Whole mount skin samples were imaged on a Zeiss LSM 880 confocal microscope with OPO using a 20x water objective. Image analysis was performed using a custom macro in FIJI. Briefly, maximum intensity z-projections of the beta-tubulin III or CGRP channel were converted to binary files that

underwent edge-detection analysis. Regions were defined by circling all stained regions. Region sizes and locations were saved.

## IHC of sectioned trigeminal ganglia

TG were dissected from 8- to 12 week old adult mice and post-fixed in 4% PFA for one hour. TG were cryo-protected overnight at 4°C in 30% sucrose-PBS, embedded in OCT, and then cryosectioned at 14 μm onto slides for staining. Slides were washed 3x in PBST (0.3% Triton X-100), blocked in 2.5% Normal Goat serum + 2.5% BSA-PBST, washed 3X in PBST, blocked in endogenous IgG block (1:10 F(ab) anti-mouse IgG (Abcam ab6668) + 1:1000 Rat anti-mouse CD16/CD32 (UCSF MAB Core) in 0.3% PBST), washed 3X in PBST and incubated overnight at 4°C in 1:1000 primary antibody in PBST + 0.5% Normal Goat Serum + 0.5% BSA. Slides were washed 3x in PBS, incubated 2 hr at RT in 1:1000 secondary antibody, washed 3X in PBS, and then incubated 30 min in 1:2000 DAPI-PBS. Slides were washed 3x in PBS and mounted in Fluoromount-G with No. 1.5 coverglass. Primary antibodies used: Mouse anti-CD45 (eBioscience #14-054-82) and Chicken anti-Peripherin (Abcam #39374). Secondary antibodies used: Goat anti-Chicken Alexa 594 (ThermoFisher #A11042) and Goat anti-Mouse Alexa 488 (Abcam #150117). DAPI (ThermoFisher #D1306) was also used to mark nuclei. Imaging of TG IHC experiments was performed on an Olympus IX71 microscope with a Lambda LS-xl light source (Sutter Instruments). For TG IHC analysis, images were analyzed using automated scripts in FIJI (ImageJ) software (*Hill, 2019*). Briefly, images were separated into the DAPI, CD45, and Peripherin channels. The minimum/maximum intensity thresholds were batch-adjusted to pre-determined levels, and adjusted images were converted to binary files. Regions were defined by circling all stained regions with pre-determined size-criteria. Region sizes and locations were saved.

## Neutrophil depletion

Neutrophils were acutely depleted using intraperitoneal injection with 250 μg aGR1 in PBS (clone RB6-8C5, a gift from D. Portnoy, UC Berkeley, or from Biolegend), 16–24 hr before behavioral and flow cytometry experiments. Depletion was verified using flow cytometry on blood collected from terminal bleed following decapitation. For longer depletion experiments using the MC903 model, mice were injected (with 250 μg aGR1 in PBS or PBS vehicle, i.p.) beginning one day prior to MC903 administration and each afternoon thereafter through day 7 of the model, or on days 8–11 for measurement of day 12 itch behaviors, and blood was collected via saphenous venipuncture at days 3, 5, or by decapitation at day eight to verify depletion.

## CXCL10 ELISA measurements in skin

Neutrophil-depleted or uninjected mice were treated with MC903 or ethanol for 7 days. On day 8, 6 mm biopsy punches of cheek skin were harvested, flash-frozen in liquid nitrogen, cryo-homogenized by mortar and pestle, and homogenized on ice for three minutes at maximum speed in 0.5 mL of the following tissue homogenization buffer (all reagents from Sigma unless stated otherwise): 100 mM Tris, pH 7.4; 150 mM NaCl, 1 mM EGTA, 1 mM EDTA, 1% Triton X-100, and 0.5% Sodium deoxycholate in ddH2O; on the day of the experiment, 200 mM fresh PMSF in 100% ethanol was added to 1 mM, with one tablet cOmplete protease inhibitor (Roche) per 50 mL, and five tablets PhosSTOP inhibitor (Roche) per 50 mL buffer. Tissues were agitated in buffer for two hours at 4°C, and centrifuged at 13,000 rpm for 20 min at 4°C. Supernatants were aliquoted and stored at −80°C for up to one week. After thawing, samples were centrifuged at 10,000 rpm for five minutes at 4°C. Protein content of skin homogenates was quantified by BCA (Thermo Scientific) and homogenates were diluted to 2 mg/mL protein in PBS and were subsequently diluted 1:2 in Reagent Diluent (R and D Systems). CXCL10 protein was quantified using the Mouse CXCL10 Duoset ELISA kit (R and D Systems; #DY466-05) according to manufacturer's instructions. Plate was read at 450 nm and CXCL10 was quantified using a seven-point standard curve (with blank and buffer controls) and fitted with a 4-parameter logistic curve.

## Acute itch behavior

Itch behavioral measurements were performed as previously described (*Shimada and LaMotte, 2008*; *Wilson et al., 2011*; *Morita et al., 2015*). Mice were shaved one week prior to itch behavior

and acclimated in behavior chambers once for thirty minutes at the same time of day on the day prior to the experiment. Behavioral experiments were performed during the day. Compounds injected: 1 µg carrier-free CXCL1 (R and D systems) in PBS, 3.31 mM AMG 487 (Tocris, prepared from 100 mM DMSO stock) in 20% HPCD-PBS, 50 mM Chloroquine diphosphate (Sigma) in PBS, along with corresponding vehicle controls. Acute pruritogens were injected using the cheek model (20 µL, subcutaneous/s.c.) of itch, as previously described (*Shimada and LaMotte, 2008*). AMG 487 (50 µL) or vehicle was injected s.c. into the rostral back skin one hour prior to recording of behavior. Behavioral scoring was performed as described above.

## Lipidomics

Skin was collected from the cheek of mice post-mortem with a 6 mm biopsy punch and immediately flash-frozen in liquid nitrogen. Lipid mediators and metabolites were quantified via liquid chromatography-tandem mass spectrometry (LC-MS/MS) as described before (*von Moltke et al., 2012*). In brief, skin was homogenized in cold methanol to stabilize lipid mediators. Deuterated internal standards ($PGE_2$-d4, $LTB_4$-d4, 15-HETE-d8, $LXA_4$-d5, DHA-d5, AA-d8) were added to samples to calculate extraction recovery. LC-MS/MS system consisted of an Agilent 1200 Series HPLC, Luna C18 column (Phenomenex, Torrance, CA, USA), and AB Sciex QTRAP 4500 mass spectrometer. Analysis was carried out in negative ion mode, and lipid 30 mediators quantified using scheduled multiple reaction monitoring (MRM) mode using four to six specific transition ions per analyte (*Sapieha et al., 2011*).

## 1-Fluoro-2,4-dinitrobenzene (DNFB) model of atopic dermatitis

The DNFB model was conducted as described previously (*Solinski et al., 2019a*). Briefly, the rostral backs of isofluorane-anesthetized mice were shaved using surgical clippers. Two days after shaving, mice were treated with 25 µL 0.5% DNFB (Sigma) dissolved in 4:1 acetone:olive oil vehicle on the rostral back using a pipette. Five days after the initial DNFB sensitization, mice were challenged with 40 µL 0.2% DNFB or 4:1 acetone:olive oil vehicle applied to the outer surface of the right ear. Twenty-four hours after DNFB or vehicle challenge, mice were euthanized and ear skin was harvested for flow cytometry.

## Statistical analyses

Different control experimental conditions (*e.g.* uninjected versus PBS-injected animals) were pooled when the appropriate statistical test showed they were not significantly different (*Supplementary file 2*). For all experiments except RNA-seq (see above), the following statistical tests were used, where appropriate: Student's t-test, one-way ANOVA with Tukey-Kramer post hoc comparison, and two-way ANOVA with Tukey Kramer or Sidak's post-hoc comparison. Bar graphs show mean ± SEM. Statistical analyses were performed using PRISM seven software (GraphPad). For all $p$ values, $*=0.01 < p<0.05$, $**=0.001 < p<0.01$, $***=0.0001 < p<0.001$, and $****=p < 0.0001$.

## Acknowledgements

The authors would like to thank members of the Bautista and Barton labs for helpful discussions on the data. We are grateful to S Ziegler (Ben Aroya Research Institute) for the gift of the TSLPR KO mouse. We also thank M Pellegrino and L Thé for pilot studies on human keratinocyte transcriptome, and T Morita and J Wong for technical assistance withTSLPR KO mouse behavioral experiments. DMB is supported by the NIH (AR059385; NS07224 and NS098097, also to RBB) and the Howard Hughes Medical Institute. GMB is supported by the NIH (AI072429, AI063302, AI104914, AI105184) and the Burroughs Wellcome Fund. JD was supported by a Long-Term Fellowship from the Human Frontier Science Program (LT-000081/2013 L). KG is supported by NIH grant EY026082. Confocal imaging experiments were conducted at the CRL Molecular Imaging Center, supported by the Helen Wills Neuroscience Institute (UC Berkeley). We would like to thank H Aaron and F Ives for their microscopy training and assistance. This work used the Functional Genomics Laboratory and the Vincent J Coates Genomics Sequencing Laboratory at UC Berkeley, supported by NIH S10 OD018174 Instrumentation Grant.

## Additional information

### Funding

| Funder | Grant reference number | Author |
|---|---|---|
| National Institute of Arthritis and Musculoskeletal and Skin Diseases | AR059385 | Diana M Bautista |
| National Institute of Neurological Disorders and Stroke | NS07224 | Rachel B Brem<br>Diana M Bautista |
| Howard Hughes Medical Institute | | Diana M Bautista |
| National Institute of Allergy and Infectious Diseases | AI072429 | Gregory M Barton |
| National Institute of Allergy and Infectious Diseases | AI063302 | Gregory M Barton |
| National Institute of Allergy and Infectious Diseases | AI104914 | Gregory M Barton |
| National Institute of Allergy and Infectious Diseases | AI105184 | Gregory M Barton |
| Burroughs Wellcome Fund | | Gregory M Barton |
| Human Frontier Science Program | LT-000081/2013-L | Jacques Deguine |
| National Eye Institute | EY026082 | Karsten Gronert |
| National Institute of Neurological Disorders and Stroke | NS098097 | Rachel B Brem<br>Diana M Bautista |

The funders had no role in study design, data collection and interpretation, or the decision to submit the work for publication.

### Author contributions

Carolyn M Walsh, Conceptualization, Formal analysis, Investigation, Visualization, Methodology, Writing—original draft, Writing—review and editing, Performed flow cytometry and behavior experiments, and made substantial contributions to Figures 1C, 1G, 2A, 2D, 2E, 2F, 2I, Figure 2—figure supplement 2, and Figure 2—figure supplement 4; Rose Z Hill, Conceptualization, Data curation, Software, Formal analysis, Investigation, Visualization, Methodology, Writing—original draft, Writing—review and editing, Performed flow cytometry, behavior, skin, somatosensory ganglia and spinal cord RNA-seq, IHC, ELISA, and made substantial contributions to Figures 1A, 1B, 1C, 1H, 1I, 1J, Figure 1—figure supplement 1, Figure 1—figure supplement 7, Figure 1—figure supplement 9, Figures 2J, 2K, 3A, 3B, 3C, 3D, 3E, 3F, 3G, 4A, 4B, 4C, 4D, 4E, 4F, 4G, Figure 4—figure supplement 1, and Figure 4—figure supplement 2; Jamie Schwendinger-Schreck, Conceptualization, Investigation, Methodology, Writing—review and editing, Performed flow cytometry, keratinocyte RNA-Seq and behavior experiments, and made substantial contributions to Figure 1—figure supplement 6, Figure 1—figure supplement 10, Figures 2A, 2E, Figure 2—figure supplement 2, and Figure 2—figure supplement 3; Jacques Deguine, Conceptualization, Formal analysis, Investigation, Methodology, Writing—review and editing, Performed flow cytometry experiments, and made substantial contributions to Figures 2A, 2B, 2C, 2E, 2G, 2H, Figure 2—figure supplement 1, Figure 2—figure supplement 2, and Figure 2—figure supplement 4; Emily C Brock, Conceptualization, Formal analysis, Investigation, Methodology, Writing—review and editing, Performed flow cytometry and behavior experiments, and made substantial contributions to Figures 1C, 1E, 1F, Figure 1—figure supplement 5, Figures 2A, 2B, 2C, 2F, 2G, 2H, Figure 2—figure supplement 1, Figures 2—figure supplement 2, and Figure 2—figure supplement 4; Natalie Kucirek, Investigation, Writing—review and editing, Performed behavior experiments, and made substantial contributions to Figures 1C and 2F; Ziad Rifi, Investigation, Writing—review and editing, Performed IHC experiments, and made substantial contributions to Figures 3C, 3D, 3E, and 3F; Jessica Wei, Investigation, Performed mass spectrometry

experiments, and made substantial contributions to Figure 1—figure supplement 10; Karsten Gronert, Resources, Supervision, Funding acquisition, Methodology, Project administration, Writing—review and editing; Rachel B Brem, Gregory M Barton, Conceptualization, Resources, Supervision, Funding acquisition, Methodology, Project administration, Writing—review and editing; Diana M Bautista, Conceptualization, Resources, Formal analysis, Supervision, Funding acquisition, Methodology, Writing—original draft, Project administration, Writing—review and editing

### Author ORCIDs
Rose Z Hill https://orcid.org/0000-0001-9558-6400
Jessica Wei https://orcid.org/0000-0002-7329-2812
Gregory M Barton https://orcid.org/0000-0002-3793-0100
Diana M Bautista https://orcid.org/0000-0002-6809-8951

### Ethics
Animal experimentation: All mice were housed in standard conditions in accordance with standards approved by the Animal Care and Use Committee of the University of California Berkeley. All experiments were performed under the policies and recommendations of the International Association for the Study of Pain and approved by the University of California Berkeley Animal Care and Use Committee (Protocol Number: 2017-02-9550).

### Decision letter and Author response
Decision letter https://doi.org/10.7554/eLife.48448.sa1
Author response https://doi.org/10.7554/eLife.48448.sa2

## Additional files

### Supplementary files
• Source data 1. The outputs of all DESeq differential expression analyses used to determine adjusted $p$ value and $\log_2$ fold change for all RNA-seq experiments in the manuscript.

• Supplementary file 1. Number of mapped reads and sample information for all RNA-seq samples represented in the manuscript.

• Supplementary file 2. Outputs of statistical tests performed on behavioral and flow cytometry data to determine whether select data sets could be combined.

• Supplementary file 3. All flow cytometry data from *Figures 1–2* represented as % of CD45[+] cells.

• Transparent reporting form

### Data availability
All data generated or analyzed during this study are included in the manuscript and supporting files. Data from RNA-seq experiments are uploaded to GEO under accession codes GSE132173 and GSE132174. Processed sequencing data (DESeq output tables) are provided as a Source data 1. Code used to analyze data is available at https://github.com/rzhill/10.1101-653873 (copy archived at https://github.com/elifesciences-publications/10.1101-653873).

The following datasets were generated:

| Author(s) | Year | Dataset title | Dataset URL | Database and Identifier |
|---|---|---|---|---|
| Bautista DM, Hill RZ | 2019 | RNA-seq of tissues from MC903- and Ethanol-treated mice | https://www.ncbi.nlm.nih.gov/geo/query/acc.cgi?acc=GSE132173 | NCBI Gene Expression Omnibus, GSE132173 |
| Bautista DM | 2019 | SLIGRL-induced gene expression changes in NHEK cells | https://www.ncbi.nlm.nih.gov/geo/query/acc.cgi?acc=GSE132174 | NCBI Gene Expression Omnibus, GSE132174 |

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
