## [Decision Letter]

Thank you for submitting your article "Neutrophils promote CXCR3-dependent itch in the development of atopic dermatitis" for consideration by *eLife*. Your article has been reviewed by three peer reviewers, and the evaluation has been overseen by Andrew King as the Senior and Reviewing Editor. The following individual involved in review of your submission has agreed to reveal their identity: Carla V Rothlin (Reviewer #1).

The reviewers have discussed the reviews with one another, and the Reviewing Editor has drafted this decision to help you prepare a revised submission.

Summary:

This paper describes a comprehensive and rigorous immunological analysis of the immune response to MC903 in mice, a model for atopic dermatitis, which affects millions of people. The authors demonstrate early neutrophilic infiltration in the affected skin, which drives initial itch responses. Furthermore, they show that neutrophil depletion leads to a significant reduction in scratching behaviour and reduced a series of hallmarks of chronic itch. They also show that neutrophils are required for the production of itch-inducing chemokines, such as CXCL10, and that pharmacological inhibition of CXCR3 (a receptor for CXCL10) inhibited itch. Overall, the reviewers thought that this is an excellent study, which provides important insights into the mechanisms underlying atopic dermatitis development and that targeting CXCL10/CXCR3 may be a promising way to treat atopic dermatitis associated itch. This paper is likely to inspire future studies on the neuroimmunological basis of chronic itch.

Essential revisions:

Although they were very positive about the study, describing the results as novel and of broad interest, the reviewers made several points that need to be addressed. These are summarized in the following, but the key points relate to the number of animals used and the variation across different time points in the study, and whether the increase in neutrophil number persists into the chronic stage of atopic dermatitis.

1) The authors propose a model wherein neutrophil infiltration drives the early response (first week), while TSLP is dispensable during the first week and only required for itch in the second week. However, the current study does not address whether neutrophils are required in the chronic setting. Do increased number of neutrophils persist in the second week? If so, does ablation of neutrophils later on affect itch behavior?

2) Are the findings broadly applicable to other models of atopic dermatitis? It would be useful to assess neutrophil recruitment in other models of atopic dermatitis.

3) Neutrophil recruitment is a common response to multiple insults, yet not all of them appear to be associated with itch. Can the authors provide an explanation for this? Is the CXCL10/CXCR3 axis selectively engaged in this condition? Or do other signals present in other settings associated with neutrophil recruitment (i.e., bacterial infection) override this pruritogenic axis? A discussion on this is merited.

4) In addition to interacting with and producing cytokines, chemokines and lipid mediators, neutrophils produce reactive oxygen and chlorinated species that, while produced to attack pathogens, can induce itch, pain and significant tissue damage. The role of this unique function of neutrophils, potentially in the residual itch responses observed by the authors, should be discussed.

5) The number of mice used in different time point varies largely. For example, in Figure 1F and Figure 1—figure supplement 2, less than 10 mice were examined for days 3 and 5 whereas around 40 mice were tested at day 8. The insignificance of the differences at days 3 and 5 is likely due to the small N numbers. In addition, large number of mice (>60/group) were used for the MC903 model which is way higher than the number of mice (around 10/group) being used in the field. What is the rationale for the large N number and the minimum number of mice needed for the model? The authors claim that neutrophils are the first immune cells to infiltrate atopic dermatitis skin. However, the data in Figure 1 suggest other immune cells (i.e. basophils in Figure 1F, inflammatory monocytes in Figure 1—figure supplement 2A, mast cells in Figure 1—figure supplement 2B) could have infiltrated atopic dermatitis skin at day 3 earlier than neutrophil recruitment at day 5. A significance increase could be found if a larger sample size is tested for these cell types at day 3 as that of day 8.

---

## [Author Response]

Essential revisions:Although they were very positive about the study, describing the results as novel and of broad interest, the reviewers made several points that need to be addressed. These are summarized in the following, but the key points relate to the number of animals used and the variation across different time points in the study, and whether the increase in neutrophil number persists into the chronic stage of atopic dermatitis.1) The authors propose a model wherein neutrophil infiltration drives the early response (first week), while TSLP is dispensable during the first week and only required for itch in the second week. However, the current study does not address whether neutrophils are required in the chronic setting. Do increased number of neutrophils persist in the second week? If so, does ablation of neutrophils later on affect itch behavior?

We thank the reviewers for suggesting we examine neutrophils in chronic itch. We have performed additional experiments assessing the presence and role of neutrophils in the second week of the MC903 model. In these new experiments we depleted neutrophils on days 8-11 to and show robust neutrophil infiltration into MC903 lesions on day 12 (Figure 2J), and loss of itch behaviors at day 12 in neutrophil-depleted mice (Figure 2K). We also performed a comprehensive review of the literature and our results are supported by a number of studies that report the presence of neutrophils in late-stage atopic dermatitis lesions. One study reported neutrophils in MC903 lesions at days 12 and 13, using histology and measurements of neutrophil-derived enzymatic activity in skin tissue (Li et al., 2017). Likewise, a study of mice deficient in filaggrin, the most commonly mutated gene in human AD patients, also found neutrophils present in the lesions of filaggrin mutant mice at 12 weeks, when the mice develop dermatitis (Saunders et al., 2016). These findings in mouse are also consistent with a number of human studies reporting the presence of neutrophils in AD (Choy et al., 2012, Mihm et al., 1976, Shalit et al., 1987). We have added the following new text to the manuscript:

Results section:

“The incomplete loss of itch behaviors on day 12 in the TSLPRKO animals (Figure 2F) raised the question of whether neutrophils might also contribute to itch during the second week of the MC903 model. […] We speculate that neutrophils and TSLP signaling comprise independent mechanisms that together account for the majority of AD itch.”

Discussion:

“Moreover, we also demonstrate that depletion of neutrophils in the second week of the MC903 model can attenuate chronic itch-evoked scratching. […] Our observations newly implicate neutrophils in setting the stage for the acute-to-chronic itch transition by triggering molecular changes necessary to develop a chronic, itchy lesion and also contributing to persistent itch.”

2) Are the findings broadly applicable to other models of atopic dermatitis? It would be useful to assess neutrophil recruitment in other models of atopic dermatitis.

Our data and others’ studies support a broad role for neutrophils in AD. At the suggestion of the reviewers, we have performed experiments measuring neutrophil infiltration in a second AD model, the 1-fluoro-2,4-dinitrobenzene (DNFB) model of AD and allergic dermatitis. A previous study showed that topical treatment with DNFB to the ear triggered chronic itch and inflammation (Solinski et al., 2019). As in the MC903 model, we found that neutrophils robustly infiltrate DNFB-treated skin (Figure 2—figure supplement 5A). We have amended the Results as follows:

“In order to ascertain whether neutrophils could be salient players in other models of AD, and not just MC903, we measured neutrophil infiltration into ear skin in the 1-fluoro-2,4dinitrobenzene (DNFB) model of atopic dermatitis, which relies on hapten-induced sensitization to drive increased IgE, mixed Th1/Th2 cytokine response, skin thickening, inflammation, and robust scratching behaviors in mice (Kitamura et al., 2018; Solinski et al., 2019; Zhang et al., 2015). Indeed, neutrophils also infiltrated DNFB- but not vehicle-treated skin (Figure 2—figure supplement 5A). […] Overall, our data support a key role for neutrophils in promoting AD itch and inflammation.”

Additionally, two studies in mouse models of AD (Li et al., 2017; Saunders et al., 2015) and several studies in humans (Choy et al., 2012; Mihm et al., 1976; Shalit et al., 1987) show that neutrophils are present in AD lesions. We also mined published transcriptomic datasets from human AD lesions for neutrophil-associated chemokines, chemokine receptors, and other transcripts. This meta-analysis yielded strong evidence supporting the presence of neutrophils in human AD lesions (Liu et al., 2019; Oetjen et al., 2017; Nattkemper et al., 2018; Suárez-Fariñas et al., 2011; Guttman-Yassky et al., 2009; Ewald et al., 2017; Li et al., 2014). We have amended the Discussion accordingly:

“In examining previous characterizations of both human and mouse models of AD and related chronic itch disorders, several studies report that neutrophils and/or neutrophil chemokines are indeed present in chronic lesions (Ewald et al., 2017; Choy et al., 2012; Guttman-Yassky et al., 2009; Suárez-Fariñas et al., 2013; Jabbari et al., 2012; Nattkemper et al., 2018; Li et al., 2017; Saunders et al., 2016; Andersson, 2014; Malik et al., 2017). Our observations newly implicate neutrophils in setting the stage for the acute-to-chronic itch transition by triggering molecular changes necessary to develop a chronic, itchy lesion and also contributing to persistent itch.”

3) Neutrophil recruitment is a common response to multiple insults, yet not all of them appear to be associated with itch. Can the authors provide an explanation for this? Is the CXCL10/CXCR3 axis selectively engaged in this condition? Or do other signals present in other settings associated with neutrophil recruitment (i.e., bacterial infection) override this pruritogenic axis? A discussion on this is merited.4) In addition to interacting with and producing cytokines, chemokines and lipid mediators, neutrophils produce reactive oxygen and chlorinated species that, while produced to attack pathogens, can induce itch, pain and significant tissue damage. The role of this unique function of neutrophils, potentially in the residual itch responses observed by the authors, should be discussed.

We agree with the reviewers that these are interesting discussion points. As such, we have added them to the manuscript:

Discussion section:

“There is a strong precedence for immune cell-neuronal interactions that drive modality-specific outcomes, such as itch versus pain, under distinct inflammatory conditions. […] It will be of great interest to the field to decipher the distinct mechanisms by which neutrophils and other immune cells interact with the nervous system to drive pain and itch.”

5) The number of mice used in different time point varies largely. For example, in Figure 1F and Figure 1—figure supplement 2, less than 10 mice were examined for days 3 and 5 whereas around 40 mice were tested at day 8. The insignificance of the differences at days 3 and 5 is likely due to the small N numbers. In addition, large number of mice (>60/group) were used for the MC903 model which is way higher than the number of mice (around 10/group) being used in the field. What is the rationale for the large N number and the minimum number of mice needed for the model? The authors claim that neutrophils are the first immune cells to infiltrate atopic dermatitis skin. However, the data in Figure 1 suggest other immune cells (i.e. basophils in Figure 1F, inflammatory monocytes in Figure 1—figure supplement 2A, mast cells in Figure 1—figure supplement 2B) could have infiltrated atopic dermatitis skin at day 3 earlier than neutrophil recruitment at day 5. A significance increase could be found if a larger sample size is tested for these cell types at day 3 as that of day 8.

We began our study by probing immune cell infiltration and itch behaviors on day 8 of the model, and later performed experiments to assess the time course of immune cell infiltration. As a result, we had many more animals for day 8 measurements than at days 3 or 5. To ensure rigor and transparency, we included all mice that were assessed using itch behavior and flow cytometry because we had no exclusion criteria defined prior to beginning the study that would justify elimination of these data. That being said, post hoc calculation of achieved power suggests that day 5 is not underpowered (see Author response table 1). Day 3 counts were quite low for all cells examined and no subtype of immune cell displayed robustly elevated counts compared to vehicle treatment (including the neutrophils, which were the most numerous subtype).

**Author response table 1. resptable1:** 

	Neutrophils	Basophils		Infl. Monos.	Mast Cells	
INPUT	Day 5	Day 8	Day 5	Day 8	Day 5	Day 8	Day 5	Day 8
Effect size	1.518	0.836	1.433	1.002	1.1711	0.8241	0.736	1.0670
α error prob	0.05	0.05	0.05	0.05	0.05	0.05	0.05	0.05
# groups	2	2	2	2	2	2	2	2
sample size	14	78	14	78	14	78	14	78
OUTPUT						
λ	32.26	54.618	28.99	102.34	19.2	52.98	7.583	31.54
Fcrit	4.747	3.966	4.747	3.966	4.747	3.966	4.747	3.966
Num. df	1	1	1	1	1	1	1	1
Denom. df	12	76	12	76	12	76	12	76
Power	0.993	1	0.986	1	0.979	0.999	0.715	0.999